# Stitch and Tell: A Structured Multimodal Data Augmentation Method for Spatial Understanding

**Hang Yin**[1], **Xiaomin He**[2], **PeiWen Yuan**[1], **Yiwei Li**[1], **Jiayi Shi**[1],
**Wenxiao Fan**[1], **Shaoxiong Feng**[3], **Kan Li**[1†]

[1] School of Computer Science, Beijing Institute of Technology
[2] School of Software and Microelectronics, Peking University
[3] Xiaohongshu Inc

{yh,peiwenyuan,liyiwei,shijiayi,wenxiaofan,likan}@bit.edu.cn
{2401210613}@stu.pku.edu.cn, {shaoxiongfeng2023}@gmail.com

## Abstract

Existing vision-language models often suffer from spatial hallucinations, i.e., generating incorrect descriptions about the relative positions of objects in an image. We argue that this problem mainly stems from the asymmetric properties between images and text. To enrich the spatial understanding ability of vision-language models, we propose a simple, annotation-free, plug-and-play method named Stitch and Tell (abbreviated as SiTe), which injects structured spatial supervision into multimodal data. It constructs stitched image–text pairs by stitching images along a spatial axis and generating spatially-aware captions or question answer pairs based on the layout of stitched image, without relying on costly advanced models or human involvement. We evaluate SiTe across three architectures including LLaVA-v1.5-7B, LLaVA-Qwen2-1.5B and HALVA-7B, two training datasets, and thirteen benchmarks. Experiments show that SiTe improves spatial understanding tasks such as $\text{MME}_{\text{Position}}$ (+5.50%) and Spatial-MM (+4.19%), while maintaining or improving performance on general vision-language benchmarks. Our findings suggest that explicitly injecting spatially-aware structure into training data offers an effective way to mitigate spatial hallucinations and improve spatial understanding, while preserving general vision-language capabilities.

## 1  Introduction

Spatial understanding, the ability to comprehend and interpret the relationships between objects in a space, is essential for tasks such as visual question answering, navigation, and embodied AI [1, 45, 40]. However, most existing vision-language models still struggle to understand and reason about spatial relationships [19, 34, 5, 39], which leads to spatial hallucination problems.

We argue that the spatial hallucination problem is primarily caused by the implicit modality gap, that is, while images contain rich multidimensional spatial features, their paired captions tend to be relatively sparse. We investigated existing large-scale multi-modal datasets, such as Conceptual Captions [28], COCO [18], VQA [1], and SBU Captions [24], and observed only a small fraction of samples contain explicit spatial information (see Table 1). As shown in Figure 1, the *spatially-aware* data means the samples whose captions include clear spatial information (e.g., "to the left of", "in front of"). During training, the model aligns visual content with limited linguistic descriptions, which may constrain its ability to capture latent spatial structures from the image alone.

This further motivates the need to introduce spatial information directly into the text. However, collecting spatial annotations through crowd-sourcing is both difficult and expensive. It often demands a carefully designed annotation interface and large-scale annotation efforts involving skilled

39th Conference on Neural Information Processing Systems (NeurIPS 2025).

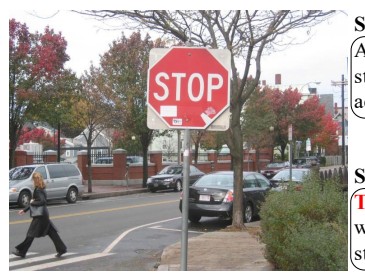

**Spatially-agnostic text :**

A stop sign stands on a quiet street, as a woman walks across the crosswalk.

**Spatially-aware text**:

**To the left of** the stop sign, a woman is walking across the street.

Figure 1: The difference between spatially-agnostic and spatially-aware text. spatially-aware text has explicit location cues that clarify object positions.

Table 1: Proportion of spatially-aware data in common datasets.

| Multimodal Datasets | Ratio |
|---|---|
| Conceptual_captions [28] | 0.0168 |
| blip_laion_cc_sbu_558K [20] | 0.0201 |
| VQA_v2 [1] | 0.0288 |
| COCO_2017 [17] | 0.0511 |
| SBU_Captions [23] | 0.0582 |
| Visual_Genome [15] | 0.0611 |
| Flickr30K [43] | 0.0732 |
| VSR [19] | 0.1898 |

annotators. One might consider leveraging data augmentation to construct spatial understanding data. However, traditional methods are not design for spatial understanding. For instance, cropping, dithering, rotation, and random erasing may break the alignment between images and text, introduce spatial noise, or even distort main semantic consistency [6, 14, 29]. Recent work has explored using large models to synthesize spatially-aware data through caption rewriting or image editing [35]. Although model-generated augmentation can enrich spatial supervision, it typically involves high computational cost and complicated processing. Given the massive scale of pretraining data, such methods are difficult to apply in practice.

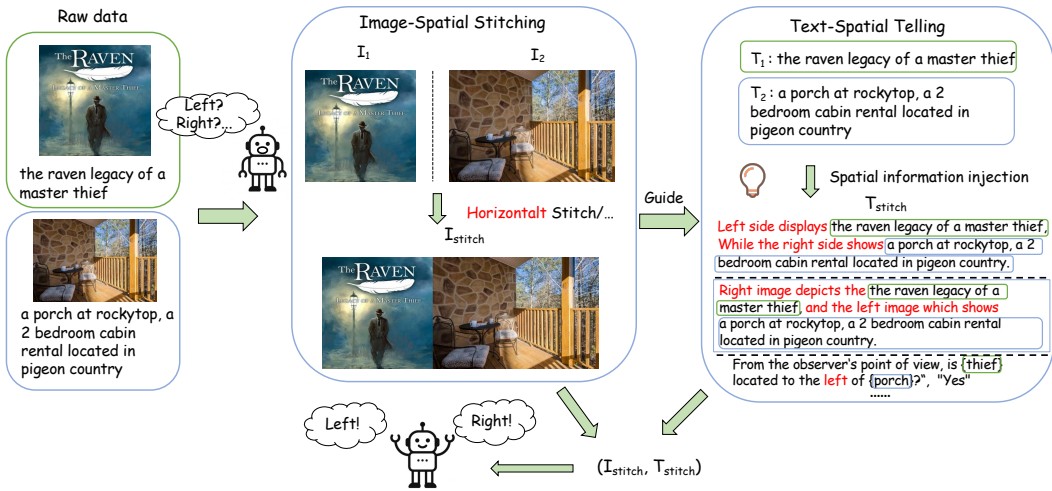

Figure 2: An Overview of Stitch and Tell

In this work, we propose a simple yet effective multi-modal data augmentation method, Stitch and Tell (SiTe), which injects spatial information into image–text pairs without relying on large-scale generation models or incurring heavy computational cost. As shown in Figure 2, SiTe consists of two main steps: IMAGE-SPATIAL STITCHING and TEXT-SPATIAL TELLING.

In the IMAGE-SPATIAL STITCHING step, we combine two images along a spatial axis (e.g., horizontal or vertical) to create a new image with an explicit spatial layout. This stitched structure naturally introduces spatial relations between regions.

In the TEXT-SPATIAL TELLING step, we generate spatially-aware textual annotations based on the original captions. First, we construct structured captions by placing the two original texts into spatial templates, such as "The right part shows $T_2$, while the left part displays $T_1$." These templates make spatial relations explicit and provide clearer supervision, helping the model associate language with visual layout. Second, we extract object nouns from each caption and generate spatial question answer (QA) pairs. For example, given "a cat" from $T_1$ and "a car" from $T_2$, we create a question like "From the observer's perspective, is the cat on the left of the car?" The answer can be automatically determined based on the stitched layout, requiring no manual labeling. These QA examples offer an

additional form of weak spatial supervision and can be directly used in instruction tuning. Compared to image-based object detection or grounding, extracting object from text is more efficient and reliable. It reduces the noise and computational overhead introduced by vision-level processing, making our method lightweight and scalable. The image–text pairs generated by SiTe introduce explicit spatial supervision through structured layout and spatially-aware text, without requiring manual annotations or model-generated rewrites. This helps bridge the gap between sparse spatial expressions in text and the richer spatial structure present in images, improving cross-modal alignment.

SiTe introduces spatial structure into multi-modal training data through simple image stitching and spatially-aware caption generation, without relying on large-scale generation models or human annotations. Each stitched sample has two image–text pairs, allowing the model to process more content per step and reducing training time by over 20% in our SiTe default setting. SiTe is easy to integrate into existing pipelines. It can be applied during both pretraining and supervised fine-tuning without modifying the model architecture.

We evaluate SiTe across three model architectures (LLaVA-v1.5-7B [20], LLaVA-Qwen2-1.5b [41] and HALVA [27]), two training dataset (558K [20] and Flickr30K), and thirteen popular benchmarks, four for spatial understanding, four for general vision-language tasks and five for more real-world benchmarks. On LLaVA-v1.5-7B, SiTe improves spatial understanding benchmarks such as $MME_{Position}$ (+5.50%) and Spatial-MM (+2.13%), while also yielding gains on general tasks like COCO-QA (+1.02%) and MMBench (+0.93%). For Qwen2-1.5B, SiTe improves MMBench by +5.33% and $MM\text{-}Vet_{Spat}$ by +4.38% in the fine-tuning stage, and achieves +10.42% on $MME_{Pos}$ and +5.01% on MMBench during pretraining. These results demonstrate that SiTe provides an effective form of weak spatial supervision that enhances spatial reasoning while maintaining competitive performance on general benchmarks, across both large and small model settings.

## 2 Related Works

**Multimodal Data Augmentation.**    To improve the generalization of vision-language models, recent work explores multimodal data augmentation strategies [13, 26]. MixGen [12] combines image interpolation with caption concatenation, but assumes semantic compatibility across samples, which may yield implausible pairs. Other approaches rely on generative models, such as StableLLaVA [16] for diffusion-based synthesis and ALIA [9] for language-guided editing. Sapkota et al. [26] categorize these methods into input mixing, caption synthesis, and adversarial perturbation. However, most require supervision, handcrafted rules, or heavy computation, limiting scalability. In contrast, we propose a structured, weakly supervised augmentation method based on spatial compositionality. It introduces explicit spatial grounding without labels or model-generated text, and integrates efficiently into standard training pipelines.

**Spatial Understanding Task**    Spatial understanding is a fundamental capability for intelligent agents to recognize and reason about the relative positions and relationships between objects. It plays a central role in real-world applications such as robotics, embodied AI, and autonomous driving [11, 21, 3]. Recent studies [30, 31, 45, 40, 42] emphasize that spatial reasoning is critical for scene understanding, navigation, and visual question answering, all of which require accurate perception of object positions and layouts. In robotics and autonomous systems, it enables agents to make informed decisions in dynamic environments [46, 2, 8, 22]. In the vision-language domain, spatial information is essential for grounding language in visual content [33, 25]. To enhance spatial reasoning, recent efforts incorporate explicit spatial features such as coordinate embeddings [4] and leverage additional modalities like depth and 3D information [7, 10]. The increasing attention to this area reflects both the challenges it poses and its importance to general intelligence, making it an active field with strong potential for future progress.

## 3 Stitch and Tell

In this section, we present Stitch and Tell, a structured multimodal data augmentation method that injects spatial knowledge by *stitching* images and *telling* their spatial layout information through structured text. We begin by introducing how images are spatially stitched to form structured visual information. We then describe how spatial relations are explicitly injected into both captions and question answer formats, allowing the model to learn spatial reasoning from weakly structured but

naturally aligned supervision. Finally, we discuss how SiTe can be effectively integrated into the model's training process.

IMAGE-SPATIAL STITCHING: Given two image–caption pairs $(I_1, T_1)$ and $(I_2, T_2)$ from the dataset, we first construct a stitched image by spatially stitching the two input images along a specific axis. Concretely, we create a blank canvas based on the larger height (for horizontal stitching) or width (for vertical stitching) of the input images, and paste $I_1$ and $I_2$ onto the canvas following a layout mode. For example, in the horizontal setting, we produce a left–right stitched image via $I_{\text{Stitch}}^{\text{LR}} = I_1 \oplus_{\text{horizontal}} I_2$. This construction introduces an explicit spatial information compaerd to the original images, helping the model acquire a grounded sense of spatial. The pseudocode of image stitching process is illustrated in Algorithm 1.

We design two image pairing strategies for constructing composite samples in the SiTe framework: (1) $\text{SiTe}_{\text{rand}}$ randomly selects two images from the dataset for horizontal concatenation, without considering their geometric proportions. This introduces diverse visual combinations but may lead to uneven scaling or excessive blank areas in the merged image. (2) $\text{SiTe}_{\text{ratio}}$ first filters vertically dominant images with a height-to-width ratio greater than 1.2, then groups them into buckets according to similar aspect ratios, and pairs images within each bucket. This ensures that the two halves of the composite image have comparable geometric structures, thereby improving spatial balance and reducing blank or redundant regions. Compared with the random pairing baseline, $\text{SiTe}_{\text{ratio}}$ increases the proportion of effective visual tokens and yields higher information density in the visual encoder's representation.

---

**Algorithm 1** Image Stitch Algorithm

1: **function** STITCH($I_1, I_2$, mode)
2: $\quad (w_1, h_1) \leftarrow$ GetSize($I_1$)
3: $\quad (w_2, h_2) \leftarrow$ GetSize($I_2$)
4: $\quad$ **if** mode = horizontal **then**
5: $\quad\quad H \leftarrow \max(h_1, h_2)$
6: $\quad\quad W \leftarrow w_1 + w_2$
7: $\quad\quad I_{\text{canvas}} \leftarrow$ NewImage($W, H$)
8: $\quad\quad$ Paste $I_1$ at $(0, 0)$ onto $I_{\text{canvas}}$
9: $\quad\quad$ Paste $I_2$ at $(w_1, 0)$ onto $I_{\text{canvas}}$
10: $\quad$ **else if** mode = vertical **then**
11: $\quad\quad W \leftarrow \max(w_1, w_2)$
12: $\quad\quad H \leftarrow h_1 + h_2$
13: $\quad\quad I_{\text{canvas}} \leftarrow$ NewImage($W, H$)
14: $\quad\quad$ Paste $I_1$ at $(0, 0)$ onto $I_{\text{canvas}}$
15: $\quad\quad$ Paste $I_2$ at $(0, h_1)$ onto $I_{\text{canvas}}$
16: $\quad$ **else**
17: $\quad\quad$ **return** Error: Invalid mode
18: $\quad$ **end if**
19: $\quad$ **return** $I_{\text{canvas}}$
20: **end function**

---

TEXT-SPATIAL TELLING. After performing image-spatial stitching, such as horizontally combining two images, we obtain a new image $I_{\text{Stitch}}^{\text{LR}}$ with an explicit spatial layout.

• **Tell the Caption.** The semantic content from the original captions $T_1$ and $T_2$ is naturally aligned with the left and right regions of the stitched image. Based on the stitching mode (e.g., left–right or top–down), we select a spatial template from repository and stitch a structured caption $T_{\text{Stitch}}$ by inserting $T_1$ and $T_2$ into the corresponding placeholders. For example, the template "*The left part shows $T_1$, and the right part displays $T_2$.*" explicitly injects spatial information through the underlined spatial phrases. These spatial information help the model associate textual semantics with the visual layout more effectively. This process does not require any additional annotation, and retains the original semantics while explicitly encoding spatial relationships, encouraging the model better align visual content with spatial language.

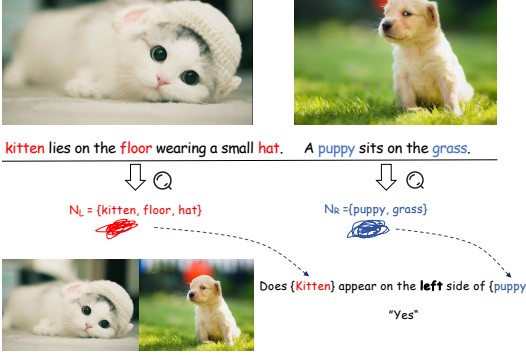

Figure 3: A construction process of spatial visual question answering data

• **Tell the Question and Answer.** Guided by the SiTe method, we further extend our method to generate spatial question–answer pairs. As shown in Figure 3, for each stitched sample, we extract noun lists from the original captions: $N_L = [o_{L,1}, \ldots, o_{L,m}]$ and $N_R = [o_{R,1}, \ldots, o_{R,n}]$, corresponding to the left and right regions of the image $I_{\text{Stitch}}^{\text{LR}}$. To reduce ambiguity, we remove

overlapping nouns shared by both sides. We then sample entity pairs $(o_L, o_R)$ from the disjoint noun sets and generate questions such as "Is the $o_L$ *on the left of* the $o_R$?"

Since the spatial position of each entity is predetermined by construction, the answer to such questions can be automatically inferred based on the object's region (e.g., $o_L \in N_L$ implies a "Yes" answer). This provides a lightweight way to generate weakly supervised spatial reasoning signals without the need for manual annotation. Compared to object detection or image-level classification, extracting entities from captions is significantly more efficient and reliable. It avoids visual ambiguity and reduces computational cost. These question-answering pairs provide the model with a lightweight but effective signal for learning spatial understanding.

The data generated by SiTe can be effectively applied across multiple stages of multimodal training. The structured captions $T_{\text{Stitch}}$ provide spatially grounded supervision during pretraining, facilitating the alignment between visual representations and spatial language without additional annotation. The constructed spatial question–answer pairs are suited for instruction tuning, where they serve as explicit prompts to strengthen the model's spatial understanding abilities. Furthermore, by modifying spatial expressions in $T_{\text{Stitch}}$ (e.g., exchanging "left" and "right"), we can generate hard negative samples that maintain the global semantics while introducing localized spatial inconsistencies. These samples can be used in contrastive learning settings to improve the model's sensitivity to spatial contradictions. Overall, SiTe offers a unified and extensible method for spatial data augmentation, supporting both generative and discriminative learning objectives with minimal supervision.

## 4 Experiment Setup

### 4.1 Models.

We conduct experiments using three representative vision-language models: LLaVA-v1.5-7B [20] (referred to as LLaVA), LLaVA-Qwen2-1.5B [41] (Qwen2), and HALVA-7B [27] (HALVA). LLaVA is built on Vicuna-7B, while Qwen2 uses Qwen2-1.5B as the language backbone. HALVA adopts a contrastive learning framework by aligning correct and hallucinated phrases at the token level for fine-grained supervision. We evaluate LLaVA and Qwen2 in both pretraining and fine-tuning stages, and apply HALVA only in the fine-tuning stage.

### 4.2 Training Sets and Augmentation Setup.

The training process of LLaVA consists of two stages: the pretraining stage and the supervised fine-tuning stage. We describe the setup for each stage separately.

**Pretraining Stage.** To evaluate the effectiveness of SiTe across different training sets, we experiment with two image–caption datasets: `blip_laion_cc_sbu_558K` (558K) and Flickr30K. As shown in Table 1, these datasets differ in the proportion of spatially informative samples, with Flickr30K containing a higher density of such spatial descriptions compared to 558K. We apply both horizontal and vertical stitching in a 1:1 ratio. By default, the stitched data make up one-third of the total set. To avoid duplicate supervision, image–caption pairs used for stitching are removed from the raw set, ensuring each image appears only once. As a result, the final number of training samples is slightly smaller than 558K. We design 35 templates for horizontal stitching and 29 templates for veritcal stitching. For each sample, a template is randomly selected from the corresponding set based on the stitching mode. This diversity in spatial phrasing encourages the model to learn spatial relations in a more flexible and robust manner, rather than relying on fixed linguistic patterns. To further compare SiTe-augmented data with existing spatial supervision data, we construct a pretraining variant by substituting part of the original image–caption data with an equal-sized spatially-focused samples from the Visual Spatial Reasoning (VSR) dataset [19]. In default setting, the number of VSR data is 5K. Additionally, we compare SiTe with two standard augmentation baselines: Rotate and Crop. For Rotate, images are randomly rotated between 0° and 360°, with captions unchanged. For Crop, 80%–100% of the image is randomly retained and cropped from a random region. Further implementation details, including the full list of spatial templates used for horizontal and vertical stitching, as well as examples of the stitched image–text pairs, are provided in Appendix B.

**Supervised Fine-Tuning Stage.** We apply the SiTe method to augment the `llava_v1_5_mix665k` dataset. Noun phrases are extracted from original sentences using `Qwen2.5-72B-Instruct` with the

prompt: *"Extract the concrete, visible physical objects or entities described in this sentence. Return a comma-separated list. Ignore abstract terms like 'type', 'color', 'time', or actions."* Based on the extracted entities, we construct spatial question–answer pairs to form a new instruction-tuning dataset. We evaluate the effect of adding 1K and 5K such samples to both LLaVA and Qwen2 models.

We further explore the way to construct spatial negative data in contrastive learning by SiTe. For each stitched caption, we generate a negative example by swapping the spatial expressions. For example, given a positive sample such as "The bottom image contains $T_1$, and the top image shows $T_2$," we construct a corresponding negative sample as "The top image contains $T_1$, and the bottom image shows $T_2$." The SiTe method simplifies this process by providing explicit spatial layouts, making it easy to create contrastive pairs based on spatial inconsistencies.

For the supervised fine-tuning stage, we define more than 20 question templates to generate spatial reasoning prompts. To reduce ambiguity, most templates explicitly incorporate the camera's perspective, using phrases such as *"**from the camera's point of view**, is left_obj located to the left of right_obj?"*to clarify directional references. See Appendix B for more details about QA templates.

SiTe provides a simple and controllable way to construct such contrastive pairs. By modifying spatial expressions in structured captions (e.g., swapping "left" and "right"), we can easily generate hallucinated variants while preserving the global semantics. In default setting for HALVA$_{\text{SiTe}}$ use 20K data, and the HALVA$_{\text{Baseline}}$ use 21.5K data.

### 4.3 Training Setup.

For the pretraining stage, we train LLaVA-v1.5-7B and LLaVA-Qwen2-1.5B using a batch size of 16 and 64 per GPU, respectively, on 8 L20Z GPUs. All pretraining experiments are conducted for 1 epoch following the original LLaVA setup. For the supervised fine-tuning stage, we use the same hardware setup and follow the original LLaVA configuration, training both models for 1 epoch. The batch sizes are set to 4, 16 and 128 per GPU for HALVA, LLaVA-v1.5-7B and LLaVA-Qwen2-1.5B, respectively. We adopt the same learning rate and weight decay as in the original model settings. In all ablation experiments, the batch size and training schedule also remain consistent, and only the ratio of stitched data is varied. For each setting, we run five times and report the average results.

### 4.4 Evaluation Setup.

We assess models in two key areas: spatial understanding and general vision-language capabilities. For spatial understanding, we use four benchmarks: COCO-QA$_{\text{Spat}}$ (subset of COCO-QA focusing on spatial questions), Spatial-MM (multiple-choice benchmark testing spatial relations between objects), MME-Position (MME subset with spatially-aware questions answered in a single word/phrase), and MM$_{\text{Spat}}$ (MM-Vet subset evaluating complex spatial reasoning). To examine whether spatial supervision affects general multimodal performance, we evaluate on COCO-QA (a multimodal dataset for basic visual understanding), VQA-v2 (diverse human-annotated QA pairs from MS-COCO for robust visual reasoning), MMBench (multi-choice benchmark covering 20 vision-language ability dimensions), and MM-Vet (open-ended and multi-choice tasks testing integrated reasoning and knowledge). To verify the model's generalization ability in spatial understanding, we conducted evaluations on CV-Bench[32] and RealWorldQA[38], both of which include questions involving 3D spatial reasoning. In addition, to explore whether this approach can also enhance model performance on high-resolution data, we carried out experiments on two high-resolution datasets, HR-Bench 8K[36] and V-Star[37]. All evaluations are conducted in a zero-shot setting on target benchmarks.

## 5 Main Results

In this section, we present the performance of SiTe and several baseline methods on both spatial understanding and general vision-language benchmarks. Notably, in the pretraining stage, the total number of training samples is kept consistent across the Baseline, Rotate, Crop, and VSR settings. For SiTe, since each stitched sample is formed by combining two image–caption pairs, the total number of training examples is reduced accordingly to ensure the model sees the same number of unique images. This means that no image–caption pair appears in both the stitched and raw data. This adjustment ensures a more fair comparison across different stitching ratios.

Table 2: Performance comparison on spatial understanding and general vision-language benchmarks with $\uparrow$ or $\downarrow$ values showing improvements or declines relative to each corresponding baseline. In the supervised fine-tuning stage, the superscript (e.g., $\text{LLaVA}_{\text{SiTe}}^{1K}$) indicates that 1K spatially-aware samples are added per stitching direction. The bottom-right corner of each model denotes the data augmentation method used. Each variant is compared to its corresponding baseline with the same backbone or data setting. *Results are using their official checkpoint.

| Model | Spatial Understanding | | | | General Vision-Language | | | |
|---|---|---|---|---|---|---|---|---|
| | $\text{COCO-QA}_{\text{Spat}}(\%)$ | Spatial-MM(%) | $\text{MME}_{\text{Pos}}$ | $\text{MM-Vet}_{\text{Spat}}$ | COCO-QA (%) | $\text{VQA}_{v2}(\%)$ | MMB(%) | MM-Vet |
| *Pretraining Stage* | | | | | | | | |
| $\text{LLaVA}_{\text{Baseline}}$ | 67.72 | 42.02 | 127.83 | 26.52 | 70.52 | 60.48 | 73.50 | 31.11 |
| $\text{LLaVA}_{\text{Rotate}}$ | 68.09 | 43.01 | 128.89 | 29.40 | 71.36 | 60.60 | 74.86 | 32.20 |
| $\text{LLaVA}_{\text{Crop}}$ | 67.82 | 42.69 | 127.78 | 28.53 | 70.93 | 60.37 | 74.41 | 31.43 |
| $\text{LLaVA}_{\text{VSR}}$ | 67.85 | 42.65 | 121.67 | 29.42 | 70.90 | 60.57 | 73.88 | 31.47 |
| $\text{LLaVA}_{\text{SiTe-rand}}$ | $68.75^{\uparrow1.03}$ | $43.78^{\uparrow1.76}$ | $133.33^{\uparrow5.50}$ | $28.43^{\uparrow1.91}$ | $71.54^{\uparrow1.02}$ | $60.53^{\uparrow0.07}$ | $74.43^{\uparrow0.93}$ | $31.40^{\uparrow0.29}$ |
| $\text{LLaVA}_{\text{SiTe-ratio}}$ | $70.06^{\uparrow2.34}$ | $44.15^{\uparrow2.13}$ | $132.80^{\uparrow4.97}$ | $28.68^{\uparrow2.16}$ | $71.19^{\uparrow0.67}$ | $60.57^{\uparrow0.09}$ | $73.64^{\uparrow0.14}$ | $32.27^{\uparrow1.16}$ |
| $\text{LLaVA}^{\text{flickr}}$ | 68.96 | 44.03 | 129.00 | 25.98 | 71.75 | 59.63 | 72.33 | 29.54 |
| $\text{LLaVA}_{\text{Rotate}}^{\text{flickr}}$ | 68.03 | 44.81 | 129.44 | 25.86 | 71.12 | 59.42 | 72.36 | 29.60 |
| $\text{LLaVA}_{\text{Crop}}^{\text{flickr}}$ | 69.34 | 42.69 | 131.11 | 28.73 | 72.12 | 60.20 | 72.68 | 29.87 |
| $\text{LLaVA}_{\text{VSR}}^{\text{flickr}}$ | 68.26 | 42.70 | 128.75 | 26.18 | 71.32 | 59.94 | 72.87 | 30.45 |
| $\text{LLaVA}_{\text{SiTe-rand}}^{\text{flickr}}$ | $71.42^{\uparrow2.46}$ | $44.97^{\uparrow0.94}$ | $130.70^{\uparrow1.70}$ | $26.20^{\uparrow0.22}$ | $73.74^{\uparrow1.99}$ | $59.73^{\uparrow0.10}$ | $72.88^{\uparrow0.55}$ | $29.59^{\uparrow0.05}$ |
| $\text{LLaVA}_{\text{SiTe-ratio}}^{\text{flickr}}$ | $71.51^{\uparrow2.55}$ | $44.31^{\uparrow0.28}$ | $131.50^{\uparrow2.50}$ | $28.60^{\uparrow2.62}$ | $73.89^{\uparrow2.14}$ | $59.94^{\uparrow0.31}$ | $71.66^{\downarrow0.67}$ | $30.97^{\uparrow1.43}$ |
| $\text{Qwen2}_{\text{Baseline}}$ | 62.25 | 40.85 | 60.75 | 20.72 | 64.72 | 53.66 | 61.67 | 22.48 |
| $\text{Qwen2}_{\text{Rotate}}$ | 60.73 | 40.78 | 60.50 | 19.62 | 60.22 | 52.77 | 66.61 | 20.97 |
| $\text{Qwen2}_{\text{Crop}}$ | 62.06 | 40.68 | 61.25 | 20.75 | 64.74 | 53.28 | 67.45 | 22.52 |
| $\text{Qwen2}_{\text{VSR}}$ | 57.18 | 41.46 | 58.50 | 22.40 | 60.22 | 54.64 | 66.58 | 22.95 |
| $\text{Qwen2}_{\text{SiTe-rand}}$ | $62.57^{\uparrow0.32}$ | $41.25^{\uparrow0.40}$ | $63.00^{\uparrow2.25}$ | $22.90^{\uparrow2.18}$ | $65.26^{\uparrow0.54}$ | $54.18^{\uparrow0.52}$ | $66.68^{\uparrow5.01}$ | $22.10^{\downarrow0.38}$ |
| $\text{Qwen2}_{\text{SiTe-ratio}}$ | $62.52^{\uparrow0.27}$ | $41.00^{\uparrow0.15}$ | $68.00^{\uparrow7.25}$ | $21.00^{\uparrow0.28}$ | $65.10^{\uparrow0.38}$ | $54.23^{\uparrow0.57}$ | $67.57^{\uparrow5.90}$ | $22.80^{\uparrow0.32}$ |
| $\text{Qwen2}^{\text{flickr}}$ | 47.10 | 38.90 | 51.25 | 10.95 | 47.90 | 42.38 | 50.90 | 10.50 |
| $\text{Qwen2}_{\text{Rotate}}^{\text{flickr}}$ | 40.08 | 39.10 | 58.33 | 9.50 | 42.45 | 39.04 | 47.86 | 9.20 |
| $\text{Qwen2}_{\text{Crop}}^{\text{flickr}}$ | 42.37 | 39.37 | 64.25 | 8.30 | 44.61 | 40.41 | 46.43 | 10.00 |
| $\text{Qwen2}_{\text{VSR}}^{\text{flickr}}$ | 46.00 | 39.90 | 67.25 | 10.15 | 48.00 | 40.93 | 48.71 | 9.60 |
| $\text{Qwen2}_{\text{SiTe-rand}}^{\text{flickr}}$ | $47.71^{\uparrow0.61}$ | $39.35^{\uparrow0.45}$ | $61.67^{\uparrow10.42}$ | $12.47^{\uparrow1.52}$ | $48.68^{\uparrow0.78}$ | $42.95^{\uparrow0.57}$ | $51.33^{\uparrow0.43}$ | $10.80^{\uparrow0.30}$ |
| $\text{Qwen2}_{\text{SiTe-ratio}}^{\text{flickr}}$ | $49.04^{\uparrow1.94}$ | $40.04^{\uparrow1.14}$ | $56.00^{\uparrow4.75}$ | $13.10^{\uparrow2.15}$ | $47.38^{\downarrow0.52}$ | $43.91^{\uparrow1.53}$ | $51.31^{\uparrow0.41}$ | $10.60^{\uparrow0.10}$ |
| *Supervised Fine-tuning Stage* | | | | | | | | |
| $\text{LLaVA}_{\text{SiTe-rand}}^{1K}$ | $68.81^{\uparrow1.09}$ | $43.16^{\uparrow1.14}$ | $128.70^{\uparrow0.87}$ | $27.06^{\uparrow0.54}$ | $71.74^{\uparrow1.22}$ | $61.17^{\uparrow0.69}$ | $74.60^{\uparrow1.10}$ | $31.20^{\uparrow0.09}$ |
| $\text{LLaVA}_{\text{SiTe-ratio}}^{1K}$ | $68.35^{\uparrow0.63}$ | $46.96^{\uparrow4.94}$ | $136.00^{\uparrow8.17}$ | $29.70^{\uparrow3.18}$ | $71.32^{\uparrow0.80}$ | $60.92^{\uparrow0.44}$ | $73.37^{\downarrow0.13}$ | $31.54^{\uparrow0.43}$ |
| $\text{LLaVA}_{\text{SiTe-rand}}^{5K}$ | $67.75^{\uparrow0.03}$ | $46.21^{\uparrow4.19}$ | $139.26^{\uparrow11.43}$ | $28.10^{\uparrow1.58}$ | $70.96^{\uparrow0.44}$ | $60.38^{\downarrow0.10}$ | $74.53^{\uparrow1.03}$ | $30.69^{\uparrow0.42}$ |
| $\text{LLaVA}_{\text{SiTe-ratio}}^{5K}$ | $68.37^{\uparrow0.65}$ | $48.58^{\uparrow6.56}$ | $141.00^{\uparrow13.17}$ | $31.05^{\uparrow4.53}$ | $70.92^{\uparrow0.40}$ | $60.82^{\uparrow0.34}$ | $73.76^{\uparrow0.26}$ | $32.15^{\uparrow1.04}$ |
| $\text{Qwen2}_{\text{SiTe-rand}}^{1K}$ | $58.53^{\downarrow3.72}$ | $41.24^{\uparrow0.39}$ | $65.33^{\uparrow4.58}$ | $23.10^{\uparrow2.38}$ | $61.77^{\downarrow2.95}$ | $54.58^{\uparrow0.92}$ | $66.37^{\uparrow4.70}$ | $23.93^{\uparrow1.45}$ |
| $\text{Qwen2}_{\text{SiTe-ratio}}^{1K}$ | $59.66^{\downarrow2.59}$ | $42.31^{\uparrow1.46}$ | $62.75^{\uparrow2.00}$ | $24.07^{\uparrow3.35}$ | $62.59^{\downarrow2.13}$ | $55.30^{\uparrow1.64}$ | $65.75^{\uparrow4.08}$ | $22.80^{\uparrow0.32}$ |
| $\text{Qwen2}_{\text{SiTe-rand}}^{5K}$ | $59.95^{\downarrow2.30}$ | $42.22^{\uparrow1.37}$ | $61.67^{\uparrow0.92}$ | $23.20^{\uparrow2.48}$ | $63.00^{\downarrow1.72}$ | $54.75^{\uparrow1.09}$ | $66.43^{\uparrow4.76}$ | $23.93^{\uparrow1.45}$ |
| $\text{Qwen2}_{\text{SiTe-ratio}}^{5K}$ | $60.22^{\downarrow2.03}$ | $41.56^{\uparrow0.71}$ | $64.50^{\uparrow3.75}$ | $25.10^{\uparrow4.38}$ | $62.86^{\downarrow1.86}$ | $54.77^{\uparrow1.11}$ | $67.00^{\uparrow5.33}$ | $24.27^{\uparrow1.79}$ |
| $\text{HALVA}_{\text{Baseline}}^{*}$ | 63.16 | 43.07 | 135.00 | 25.70 | 67.12 | 61.67 | 72.44 | 30.00 |
| $\text{HALVA}_{\text{SiTe}}$ | $64.77^{\uparrow1.61}$ | $44.15^{\uparrow1.08}$ | $123.33^{\downarrow11.67}$ | $26.10^{\uparrow0.40}$ | $68.54^{\uparrow1.42}$ | $61.03^{\downarrow0.64}$ | $71.54^{\downarrow0.90}$ | $30.80^{\uparrow0.80}$ |

## 5.1 Evaluation on Spatial Understanding Benchmarks

We first evaluate the impact of SiTe on spatial understanding. The results are provided in Table 2. In pretraining stage, experiments are conducted under the pretraining setting using two datasets (558K and Flickr30K) and two backbones: LLaVA-v1.5-7B and LLaVA-Qwen2-1.5B. We compare SiTe against baseline training, traditional augmentations (Rotate, Crop), and training data mixed with VSR data.

**Effect of SiTe-Augmented Pretraining.** SiTe-augmented method consistently improves spatial understanding performance across all benchmarks and model backbones. This gain can be attributed to its explicit introduction of structured spatial knowledge during pretraining. By stitching two images with corresponding spatial information (e.g., "left to", "top of"), the model learns to associate language not only with object content, but also with relative layout.

The $\text{LLaVA}_{\text{Rotate}}$ model shows moderate improvement on the **558K** dataset. This may be because spatial expressions are relatively sparse in 558K, so rotating the image does not heavily conflict with the caption. In some cases, it may even help the model become more robust by learning to generalize

across different viewpoints. However, when using Flickr30K as the training set, where captions contain richer spatial descriptions, LLaVA$_{\text{Rotate}}^{\text{flickr}}$ performs worse than LLaVA$^{\text{flickr}}$ on several spatial benchmarks. For instance, on COCO-QA$_{\text{Spat}}$, accuracy drops by 0.93%. This suggests that rotation may introduce misalignment between spatial phrases in the caption and the actual image layout, potentially resulting in implicit negative supervision. And the performance of the Crop baseline is less stable. Random cropping may inadvertently remove key objects or semantic regions, leading to weakened image-text alignment. Although the retained area is still between 0.8 to 1, the risk of disrupting mismatch grounding remains high—potentially introducing merrors where the caption does not accurately reflects the image content.

Among SiTe variants, SiTe-rand corresponds to the original random pairing strategy, while SiTe-ratio adopts a ratio-based image pairing scheme that aligns images by aspect ratio before stitching. Compared with SiTe-rand, SiTe-ratio achieves further gains on most spatial benchmarks, indicating that more efficient image composition improves the performance of spatial supervision. The random variant still provides strong overall enhancement across models and datasets, showing that SiTe is robust even without additional pairing constraints.

Flickr30K contains higher proportion of spatially-aware data, which helps allows LLaVA$_{\text{SiTe}}^{\text{flickr}}$ learn both spatial relations within each image and layout patterns from stitched pairs, and perform better than LLaVA$_{\text{SiTe}}$(from 68.75% to 71.42%). This allows the model to better understand complex spatial structures, even with less data in pretraining, and achieve strong overall performance.

**Effect of SiTe-Augmented Supervised Fine-Tuning.** SiTe continues to deliver strong results during the supervised fine-tuning stage. By adding 1K and 5K spatially-aware samples per stitching mode to the instruction tuning set, we observe consistent improvements across most spatial benchmarks. For example, after adding 5K horizontal and 5K vertical QA samples to LLaVA-v1.5-7B in fine-tuning stage, the accuracy on Spatial-MM increases from 42.97% to 46.21%, outperforming the baseline by 4.19%. These gains come from our data construction strategy, which extracts noun entities from stitched regions and generates spatial QA instructions from the camera's perspective. This enables the model to learn spatial reasoning patterns from naturally aligned weak supervision.

The SiTe-augmented method can also be used to construct contrastive examples. We apply spatially stitched text to HALVA by replacing only spatial information (e.g., "left" $\leftrightarrow$ "right") while keeping the other semantics information unchanged. In contrast, the HALVA method modifies objects, relations, and attributes. Despite this simpler setup, HALVA$_{\text{SiTe}}$ still outperforms the HALVA$_{\text{Baseline}}$ on most benchmarks.

We observe a slight performance drop for Qwen2$_{\text{SiTe}}$ on COCO-QA. This suggests a potential unalignment when fine-tuning small-capacity models on highly structured spatial QA data. Specifically, SiTe provides binary Yes/No questions focused spatial understanding, which differ in format and answer space from COCO-QA's diverse and open-ended questions. As a result, the model may over-adapt to the binary QA style, leading to reduced flexibility when answering questions that require generative reasoning or retrieval of specific object names. This effect is more pronounced in smaller models like Qwen2-1.5B, which are more sensitive to supervision bias and have limited generalization capacity.

## 5.2    Evaluation on General Vision-Language Benchmarks

We further evaluate whether SiTe compromises general vision-language capabilities while improving spatial understanding. Results are summarized in Table 2. For models based on LLaVA-v1.5-7B, SiTe consistently maintains or improves general performance. Notably, SiTe-augmented in LLaVA maintains consistent improvements on general vision-language benchmarks during the pretraining stage. This suggests that introducing spatial understanding at this stage enhances the model's ability to generalize to out-of-distribution (OOD) scenarios. In the supervised fine-tuning stage stage, SiTe still shows mostly positive gains, though we observe slight drops on VQA-v2 and MM-Vet. This may be due to the nature of instruction tuning, where a large number of spatial-focused examples shift the model's preference, potentially affecting its performance on general tasks. For Qwen2, which has relatively limited capacity, SiTe also brings clear benefits in pretraining. In particular, Qwen2$_{\text{SiTe}}$ improves MMBench by 5.01%. However, during supervised fine-tuning stage, Qwen2 shows a slight decrease on COCO-QA, indicating that instruction tuning tends to reinforce domain-specific behavior. For smaller models, this may reduce the ability to follow diverse instructions, leading to weaker

generalization. These results further highlight the importance of spatial supervision during pretraining for improving downstream robustness.

We further conducted experiments on both high-resolution and 3D spatial benchmarks to evaluate the generalization capability of our approach. Specifically, we tested models enhanced with SiTe on HR-Bench 8K and V-Star, where they consistently achieved higher scores than the baselines, demonstrating improved perception of fine-grained visual details.

To assess spatial reasoning in more realistic and complex settings, we also evaluated the models on CV-Bench and RealWorldQA, which include diverse questions involving 3D spatial relationships. The results show that injecting structured spatial supervision through SiTe significantly improves performance on out-of-distribution spatial reasoning tasks and generalizes well to more challenging real-world scenarios. We hypothesize that these gains arise from the model's enhanced ability to represent fundamental directional relations, which in turn supports broader spatial understanding across different dimensions.

Table 3: Performance comparison across real-world and high resolution benchmarks.

| Model | CV-Bench 2D (%) | CV-Bench 3D (%) | RealworldQA (%) | HRBench-8K (%) | V-Star (%) |
|---|---|---|---|---|---|
| LLaVA$_{Baseline}$ | 52.56 | 33.43 | 55.12 | 30.90 | 48.34 |
| **LLaVA$_{SiTe-rand}$** | 54.28$^{\uparrow 1.72}$ | 35.50$^{\uparrow 2.07}$ | 55.21$^{\uparrow 0.09}$ | 32.56$^{\uparrow 1.66}$ | 49.95$^{\uparrow 1.61}$ |
| **LLaVA$_{SiTe-ratio}$** | 55.58$^{\uparrow 3.02}$ | 36.53$^{\uparrow 3.10}$ | 55.26$^{\uparrow 0.14}$ | 33.72$^{\uparrow 2.82}$ | 50.13$^{\uparrow 1.79}$ |
| LLaVA$^{flickr}$ | 52.87 | 28.39 | 54.51 | 33.52 | 49.32 |
| **LLaVA$^{flickr}_{SiTe-rand}$** | 54.06$^{\uparrow 1.19}$ | 38.66$^{\uparrow 10.27}$ | 55.28$^{\uparrow 0.77}$ | 34.53$^{\uparrow 1.01}$ | 47.28$^{\downarrow 2.04}$ |
| **LLaVA$^{flickr}_{SiTe-ratio}$** | 55.07$^{\uparrow 2.20}$ | 39.80$^{\uparrow 11.41}$ | 56.25$^{\uparrow 1.74}$ | 33.94$^{\uparrow 0.42}$ | 50.79$^{\uparrow 1.47}$ |
| Qwen2$_{Baseline}$ | 46.77 | 47.63 | 52.28 | 32.25 | 37.83 |
| **Qwen2$_{SiTe-rand}$** | 48.16$^{\uparrow 1.39}$ | 50.36$^{\uparrow 2.73}$ | 54.44$^{\uparrow 2.16}$ | 32.67$^{\uparrow 0.42}$ | 38.92$^{\uparrow 1.09}$ |
| **Qwen2$_{SiTe-ratio}$** | 47.47$^{\uparrow 0.70}$ | 50.40$^{\uparrow 2.77}$ | 52.94$^{\uparrow 0.66}$ | 32.94$^{\uparrow 0.69}$ | 38.09$^{\uparrow 0.26}$ |
| Qwen2$^{flickr}$ | 42.59 | 51.57 | 43.20 | 32.63 | 36.13 |
| **Qwen2$^{flickr}_{SiTe-rand}$** | 43.64$^{\uparrow 1.05}$ | 51.67$^{\uparrow 0.10}$ | 42.75$^{\downarrow 0.45}$ | 33.34$^{\uparrow 0.71}$ | 36.52$^{\uparrow 0.39}$ |
| **Qwen2$^{flickr}_{SiTe-ratio}$** | 44.44$^{\uparrow 1.85}$ | 51.50$^{\downarrow 0.07}$ | 49.74$^{\uparrow 6.54}$ | 34.33$^{\uparrow 1.70}$ | 36.39$^{\uparrow 0.26}$ |

We also calculate the computational time efficiency improvement brought about by pre-training training using SiTe method. Taking default as an example, due to the decrease in the number of data strips, the time to run an epoch in the pre-training stage is 77.4% of the baseline.

In summary, Stitch and Tell effectively enhances spatial supervision by increasing data density through structured augmentation, without sacrificing the original knowledge. This approach leads to stronger spatial understanding and remains competitive on general vision-language benchmarks, demonstrating its broad applicability.

## 5.3 Qualitative Analysis

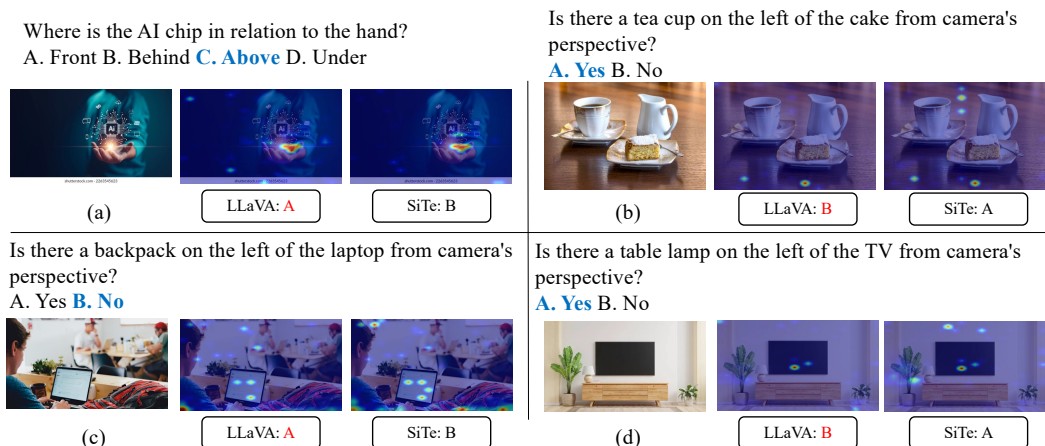

Figure 4: Qualitative comparisons between LLaVA and SiTe. SiTe can better distribute attention between objects and judge their spatial relationships.

We visualize attention differences MLLM-Know [44], which highlights image regions where the model assigns more attention under task-specific prompts compared to general prompts. A qualitative

comparison between LLaVA-v1.5-7B and SiTe is shown in Figure 4. In Figure 4 (a), when asked about the relation between the AI chip and the hand, LLaVA mainly attends to the hand while neglecting the chip, resulting in an incorrect answer. In contrast, SiTe correctly focuses on both entities. In Figure 4 (b), where the question involves the left side of the cake, SiTe allocates more attention to the region between the cup and the cake, effectively capturing their spatial arrangement. Similarly, in Figure 4 (c) and (d), SiTe exhibits more precise attention distribution across the referenced objects, which contributes to more accurate spatial understanding.

## 5.4 Ablations

Recalling the SiTe method in the pretraining stage, we define a stitching ratio parameter $\lambda$ to measure the proportion between stitched and raw samples in the final training set. Let $T$ denote the number of original image–caption pairs, and $N$ be the number of stitched samples constructed for each stitching mode (e.g., horizontal or vertical). Since each stitched sample is formed by consuming two raw samples, the total number of stitched samples is $2N$, and the remaining raw examples become $T - 4N$, assuming two stitching modes are used. Let $\mathcal{D}_{\text{Stitch}}$ and $\mathcal{D}_{\text{Raw}}$ denote the sets of stitched and retained raw examples, respectively. The final ratio $\lambda$ is defined as: $\lambda = \frac{|\mathcal{D}_{\text{Stitch}}|}{|\mathcal{D}_{\text{raw}}|} = \frac{2N}{T-4N}$.

We present an ablation study using the 558K dataset for pretraining. Specifically, we introduce seven SeTi variants with different stitching ratios, as summarized in Table 5. We observe that when the stitching ratio $\lambda$ is around 1/3, the model achieves overall strong performance on spatial understanding benchmarks while maintaining competitive results on general vision-language tasks. This setting provides a favorable trade-off between spatial supervision and overall data diversity. A similar trend is observed when using the Flickr30K dataset for training, and the more detail ablation results and analysis are included in the Appendix C.

Table 4: Settings of SiTe-rand as variants of SiTe$_{\text{default}}$ used for the ablation study during pretraining on the 558K dataset. Each variant adopts a different ratio $\lambda$, representing the proportion of stitched samples to the remaining raw samples in the training set, and the total number of training instances varies accordingly.

| Setting | Images(Total data size) | $\lambda$ |
|---|---|---|
| **SiTe$_{\text{default}}$** | 458K | 1 : 3.58 |
| (a) N=1K | 556K | 1 : 277.0 |
| (b) N=5K | 548K | 1 : 53.8 |
| (c) N=10K | 538K | 1 : 25.9 |
| (d) N=100K | 358K | 1 : 0.79 |
| (e) N=139K | 280K | 1 : 0.01 |

Table 5: Performance on spatial understanding benchmarks (left) and general vision-language benchmarks (right) under different spatial data augmentation settings.

| Setting | Spatial Understanding | | | | General Vision-Language | | | |
|---|---|---|---|---|---|---|---|---|
| | COCO-QA$_{\text{Spatial}}$ | Spatial-MM | MME$_{\text{Position}}$ | MM-Vet$_{\text{Spat}}$ | COCO-QA | VQA$_{\text{v2}}$ | MMB | MM-Vet |
| **SiTe$_{\text{default}}$** | 68.75 | **43.78** | **133.33** | **28.43** | 74.43 | 60.53 | 74.43 | **31.40** |
| (a) | 69.23 | 42.90 | 126.3 | 27.54 | 74.05 | 60.50 | 74.05 | 30.58 |
| (b) | 67.58 | 42.42 | 132.67 | 27.59 | 74.11 | 60.35 | 74.11 | 31.34 |
| (c) | 68.13 | 42.83 | 128.67 | 27.50 | 74.04 | 60.40 | 74.04 | 31.45 |
| (d) | 68.83 | 43.10 | 132.17 | 27.63 | 71.53 | **60.55** | 74.55 | 31.27 |
| (e) | **71.09** | 43.72 | 132.50 | 28.06 | **74.74** | 60.32 | **74.74** | 31.29 |

## 6 Conclusion

In this work, we present Stitch and Tell, a simple and scalable data augmentation strategy that injects spatial structure into vision-language training. SiTe combines image stitching with spatially-aware caption generation to provide weak spatial supervision without requiring human annotation or architectural changes. We apply SiTe to multiple backbones, including LLaVA, Qwen2 and HALVA, across two training datasets and thirteen benchmarks. Experiments show consistent improvements on both spatial understanding and general vision-language tasks. These results demonstrate that encoding spatial structure into training data can improve cross-modal alignment and spatial reasoning, while maintaining strong general performance. We hope this work provides a lightweight and broadly applicable approach to structured multimodal data augmentation for spatial understanding.

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

# A    Limitation

While Stitch and Tell demonstrates strong performance and scalability across multiple settings, it currently focuses on simple binary spatial configurations—primarily horizontal (left-right) and vertical (top-down) compositions. This design enables efficient supervision generation and robust model alignment, but may limit the model's exposure to more complex spatial relationships that occur in real-world environments, such as diagonal layouts, circular arrangements, or relative distances in 3D space. We believe extending SiTe to handle richer spatial topologies, possibly by incorporating depth maps, multi-image compositions, or 3D-aware stitching strategies, is a promising direction for future work.

# B    Data Template

To support structured data generation in our Stitch and Tell method, we employ a collection of spatially guided templates that are automatically generated and manually filtered for diversity and clarity by GPT-4o. These templates are designed to inject explicit spatial structure into image–text data and are categorized as follows:

- **Spatial Caption Templates:** As shown in Table 6 and Table 7, we collect 35 templates for horizontal stitching and 29 for vertical stitching. These templates are automatically produced using a language model and then curated to ensure naturalness and correctness. Each template organizes two independent captions into a single sentence with clear spatial cues, such as *"Left side displays: {caption1}, Right side shows: {caption2}"*. The diversity of expressions encourages the model to learn generalized spatial grounding rather than relying on specific lexical patterns.
- **Spatial QA Templates:** As shown in Table 8 and Table 9, we generate 20 question templates in each stitching way (horizontal or vertical). These templates are used to form weakly supervised question answer pairs such as *"From the observer's viewpoint, is the cat on the left of the car?"*, with answers derived directly from the known image layout. Templates were automatically generated with instruction-tuned models and then filtered to preserve clarity, directional accuracy, and grammatical diversity.

Table 6: Caption templates used for constructing structured spatial descriptions in S&T. Placeholders {caption1} and {caption2} represent the original descriptions from the left and right images, respectively.

| No. | Template |
|-----|----------|
| 1. | On the left, {caption1}. Meanwhile, the right side presents {caption2}. |
| 2. | This composite image showcases {caption1} on the left, contrasting beautifully with {caption2} on the right. |
| 3. | The left section highlights {caption1}, while the right side draws attention to {caption2}. |
| 4. | Displayed in the left half is {caption1}, while the right half features {caption2}. |
| 5. | {caption1} on the left and {caption2} on the right. |
| 6. | Left image shows: {caption1} \| Right image shows: {caption2}. |
| 7. | On the left: {caption1} \|\| On the right: {caption2}. |
| 8. | First image description: {caption1}, Second image description: {caption2}. |
| 9. | Image pair – Left: {caption1} ; Right: {caption2}. |
| 10. | [Left] {caption1} [Right] {caption2}. |
| 11. | Left side displays: {caption1} <==> Right side displays: {caption2}. |
| 12. | Left view: {caption1} — Right view: {caption2}. |
| 13. | Left portion: {caption1} » Right portion: {caption2}. |
| 14. | Left panel shows: {caption1} \|\| Right panel shows: {caption2}. |
| 15. | Left section: {caption1} <-> Right section: {caption2}. |
| 16. | The right one shows {caption2}, while the left displays {caption1}. |
| 17. | A pair of images: on the right we see {caption2}, and on the left there's {caption1}. |
| 18. | The right image contains {caption2}, paired with a left image showing {caption1}. |
| 19. | Two scenes: {caption2} on the right, accompanied by {caption1} on the left. |
| 20. | Right image depicts {caption2}, contrasting with the left image showing {caption1}. |
| 21. | {caption1} (left) vs {caption2} (right). |
| 22. | Contrast in perspective – left presents {caption1}, while right shows {caption2}. |
| 23. | Visual contrast: Left shows {caption1}, Right shows {caption2}. |
| 24. | Left portrays {caption1}, Right highlights {caption2}. |
| 25. | Left vs Right: {caption1} & {caption2}. |
| 26. | Left illustrates {caption1}, whereas right {caption2}. |
| 27. | Left image showcasing {caption1}, and right image featuring {caption2}. |
| 28. | {caption1} and {caption2}. |
| 29. | {caption1}, {caption2}. |
| 30. | Left: {caption1} Right: {caption2}. |
| 31. | {caption1} {caption2}. |
| 32. | Left/Right: {caption1}/ {caption2}. |
| 33. | \| {caption1} \| vs \| {caption2} \| |

Table 7: Templates used for generating structured captions in top–down stitched images. Placeholders {caption1} and {caption2} represent captions from the top and bottom image regions.

| No. | Caption Template |
| --- | --- |
| 1. | On the top, you can see {caption1}, the bottom side presents {caption2}. |
| 2. | This composite image showcases {caption1} on the top, with {caption2} on the bottom. |
| 3. | The top section highlights {caption1}, while the bottom side draws attention to {caption2}, creating an engaging visual comparison. |
| 4. | Displayed in the upper half is {caption1}, while the lower half features {caption2}, illustrating unique attributes side by side. |
| 5. | A striking juxtaposition: {caption1} on the top and {caption2} on the bottom, offering an interesting visual narrative. |
| 6. | Top image shows: {caption1}, Bottom image shows: {caption2}. |
| 7. | On the top: {caption1}. On the bottom: {caption2}. |
| 8. | First image description: {caption1}, Second image description: {caption2}. |
| 9. | Top: {caption1}; Bottom: {caption2}. |
| 10. | Top part display: {caption1} Bottom part display: {caption2}. |
| 11. | Upper displays: {caption1}, Lower displays: {caption2}. |
| 12. | Top view: {caption1}, and Bottom view: {caption2}. |
| 13. | Top portion: {caption1} » Bottom portion: {caption2}. |
| 14. | Upper panel shows: {caption1} ‖ Lower panel shows: {caption2}. |
| 15. | Upper section: {caption1} <-> Lower section: {caption2}. |
| 16. | The bottom one shows {caption2}, while the top displays {caption1}. |
| 17. | A pair of images: on the bottom we see {caption2}, and on the top there's {caption1}. |
| 18. | The bottom image contains {caption2}, the top image showing {caption1}. |
| 19. | Two scenes: {caption2} on the lower side, accompanied by {caption1} on the upper. |
| 20. | Bottom image depicts {caption2}, contrasting with the top image which shows {caption1}. |
| 21. | {caption1} (top) vs {caption2} (bottom). |
| 22. | Top side presents {caption1}, while bottom side shows {caption2}. |
| 23. | Top shows {caption1}, Bottom shows {caption2}. |
| 24. | Top portrays {caption1}, Bottom highlights {caption2}. |
| 25. | Top vs Bottom: {caption1} & {caption2}. |
| 26. | Top illustrates {caption1}, whereas bottom captures {caption2} in detail. |
| 27. | A split view: top image showcasing {caption1}, and bottom image featuring {caption2}. |
| 28. | {caption1} and {caption2}. |
| 29. | {caption1}, {caption2}. |
| 30. | Top: {caption1}, Bottom: {caption2}. |

Table 8: List of spatial QA templates used in instruction tuning. Each template is instantiated with {left_obj} and {right_obj}, and paired with a binary answer.

| ID | Template (with placeholders) | Answer |
|---|---|---|
| 1. | From the observer's point of view, is {left_obj} located to the left of {right_obj}? | Yes |
| 2. | From the camera's viewpoint, can we see {right_obj} on the right of {left_obj}? | Yes |
| 3. | Looking from the front, is {left_obj} placed to the right of {right_obj}? | No |
| 4. | As observed from the viewer's perspective, does {right_obj} appear left of {left_obj}? | No |
| 5. | From the point of view of the observer, is {left_obj} on the left side of {right_obj}? | Yes |
| 6. | Is {right_obj}, from the camera's perspective, situated to the right of {left_obj}? | Yes |
| 7. | When facing the image, does the left contain {left_obj} and the right contain {right_obj}? | Yes |
| 8. | From a frontal viewpoint, is {left_obj} to the left side of {right_obj}? | Yes |
| 9. | As seen in the combined image, is {right_obj} located right of {left_obj}? | Yes |
| 10. | To the viewer, does {left_obj} appear on the right side of {right_obj}? | No |
| 11. | Does {left_obj} appear on the left side of {right_obj}? | Yes |
| 12. | Is {right_obj} located to the left of {left_obj}? | No |
| 13. | Can {left_obj} be found on the right of {right_obj}? | No |
| 14. | Is {right_obj} positioned on the right of {left_obj}? | Yes |
| 15. | Would you say {left_obj} is left to {right_obj}? | Yes |
| 16. | Is {right_obj} on the left of {left_obj} in this composition? | No |
| 17. | Does the image on the left contain {left_obj} while the right image contains {right_obj}? | Yes |
| 18. | Is {left_obj} positioned on the right of {right_obj} instead? | No |
| 19. | In this pair, is {right_obj} located to the right of {left_obj}? | Yes |
| 20. | If you observe carefully, is {left_obj} on the right and {right_obj} on the left? | No |

Table 9: Templates for generating top–down spatial question–answer (QA) pairs. Placeholders {top_obj} and {bottom_obj} denote entities from the top and bottom image regions.

| No. | QA Template | Answer |
|---|---|---|
| 1. | In the image, is {top_obj} located above {bottom_obj}? | Yes |
| 2. | From the viewpoint of the observer, is {bottom_obj} below {top_obj}? | Yes |
| 3. | Would you say that {top_obj} is placed underneath {bottom_obj}? | No |
| 4. | As observed from the front, is {bottom_obj} situated on top of {top_obj}? | No |
| 5. | Does the top part of the image contain {top_obj} while the bottom part has {bottom_obj}? | Yes |
| 6. | Looking from top to bottom, do you first see {top_obj}, then {bottom_obj}? | Yes |
| 7. | Is {bottom_obj} placed above {top_obj} in this composition? | No |
| 8. | From top-down perspective, is {top_obj} above {bottom_obj}? | Yes |
| 9. | In this combined image, does {top_obj} appear at the top and {bottom_obj} at the bottom? | Yes |
| 10. | Is {top_obj} below {bottom_obj}? | No |
| 11. | Would you say {bottom_obj} is at a lower vertical position than {top_obj}? | Yes |
| 12. | Does the vertical layout place {top_obj} higher than {bottom_obj}? | Yes |
| 13. | Is {bottom_obj} appearing above {top_obj} in this image? | No |
| 14. | Do you see {top_obj} on the upper half and {bottom_obj} on the lower half of the image? | Yes |
| 15. | Can we find {bottom_obj} positioned higher than {top_obj} in the image? | No |
| 16. | In the vertical layout, is {top_obj} stacked above {bottom_obj}? | Yes |
| 17. | Does {top_obj} sit at the bottom while {bottom_obj} is on top? | No |
| 18. | Is the object {top_obj} visually located above {bottom_obj} from this angle? | Yes |
| 19. | Would you agree that {bottom_obj} is beneath {top_obj} in this view? | Yes |
| 20. | Is {bottom_obj} placed at the upper portion of the image, above {top_obj}? | No |

## C  Additional Experiments and Results

### C.1  Ablation study

Table 11: Performance of LLaVA-Qwen2-1.5B pretrained on the 558K dataset under different spatial data augmentation strategies. Results are reported on spatial understanding benchmarks (left) and general vision-language benchmarks (right).

| Setting | Spatial Understanding | | | | General Vision-Language | | | |
|---|---|---|---|---|---|---|---|---|
| | COCO-QA$_{Spatial}$ | Spatial-MM | MME$_{Position}$ | MM-Vet$_{Spat}$ | COCO-QA | VQA$_{v2}$ | MMB | MM-Vet |
| SiTe$_{default}$ | **62.57** | **41.25** | 63.00 | **22.90** | **65.26** | **54.18** | 66.68 | **22.10** |
| (c) | 60.88 | 41.08 | 63.20 | 20.80 | 64.81 | 53.50 | 67.51 | 21.04 |
| (d) | 59.83 | 41.11 | **64.00** | 21.75 | 62.74 | 52.83 | **70.32** | 21.30 |
| (e) | 53.20 | 39.66 | 55.00 | 18.15 | 54.74 | 46.80 | 64.47 | 17.10 |

Table 12: Performance of LLaVA-v1.5-7B pretrained on the Flickr30K dataset under different spatial data augmentation strategies. Results are reported on spatial understanding benchmarks (left) and general vision-language benchmarks (right).

| Setting | Spatial Understanding | | | | General Vision-Language | | | |
|---|---|---|---|---|---|---|---|---|
| | COCO-QA$_{Spatial}$ | Spatial-MM | MME$_{Position}$ | MM-Vet$_{Spat}$ | COCO-QA | VQA$_{v2}$ | MMB | MM-Vet |
| SiTe$_{default}$ | **71.42** | **44.97** | **130.70** | 26.20 | **73.74** | **59.73** | **72.88** | 29.59 |
| (1) | 69.80 | 42.96 | 130.30 | 27.54 | 72.53 | 59.65 | 72.56 | **31.06** |
| (2) | 69.63 | 44.50 | 126.90 | **27.59** | 72.44 | 59.10 | 72.42 | 29.40 |
| (3) | 71.03 | 44.21 | 133.00 | 26.17 | 73.28 | 58.73 | 72.31 | 29.19 |

Table 13: Performance of LLaVA-Qwen2-1.5B pretrained on the Flickr30K dataset under different spatial data augmentation strategies. Results are reported on spatial understanding benchmarks (left) and general vision-language benchmarks (right).

| Setting | Spatial Understanding | | | | General Vision-Language | | | |
|---|---|---|---|---|---|---|---|---|
| | COCO-QA$_{Spatial}$ | Spatial-MM | MME$_{Position}$ | MM-Vet$_{Spat}$ | COCO-QA | VQA$_{v2}$ | MMB | MM-Vet |
| SiTe$_{default}$ | 47.71 | 39.35 | **61.00** | 12.47 | 48.68 | **42.95** | 51.33 | 10.80 |
| (1) | 47.57 | 39.28 | 57.60 | 12.00 | 48.17 | 40.78 | 49.43 | 10.54 |
| (2) | **48.16** | 38.88 | 56.50 | **12.60** | **48.76** | 41.69 | **52.04** | **11.23** |
| (3) | 43.06 | **39.63** | 60.00 | 11.65 | 45.72 | 39.36 | 49.09 | 10.53 |

To identify the optimal stitching ratio for spatial data augmentation, we perform ablation experiments on four settings: LLaVA-v1.5-7B trained on 558K and Flickr30K, Qwen2-1.5B trained on 558K and Flickr30K. In each setting, we vary the number of stitched samples and report performance on both spatial understanding and general vision-language benchmarks.

In the main text, we present ablation results of our SiTe method on LLaVA using the 558K dataset. Here, we provide additional ablations on two new settings: LLaVA-v1.5-7B pretrained on Flickr30K, and LLaVA-Qwen2-1.5B pretrained on both 558K and Flickr30K.

Table 10: Settings of SiTe variants used for ablation study during pretraining on the Flickr30K dataset. Each variant uses a different ratio $\lambda$, which denotes the proportion of stitched samples to the remaining raw samples in the training set. The total number of training data varies accordingly.

| Setting | Images(Total data size) | $\lambda$ |
|---|---|---|
| SiTe$_{default}$ | 24K | 1 : 3.0 |
| (1) N=1K | 28K | 1 : 13.0 |
| (2) N=5K | 20K | 1 : 1.0 |
| (3) N=7K | 16K | 1 : 0.14 |

For the 558K dataset, we evaluate Qwen2 under four configurations with different stitching sizes: $N = 10K$ (setting (c)), $N = 50K$ (default setting on 558K), $N = 100K$ (setting (d)), and $N = 139K$ (setting (e)). The corresponding results are summarized in Table 11.

For Flickr30K, we evaluate on $N = 1$K (setting (1)), $N = 3$K (default setting on `Flickr30K`), $N = 5$K (setting (2)), and $N = 7$K (setting (3)). Dataset statistics are shown in Table 10, and the results are reported in Table 12 and Table 13 for LLaVA and Qwen2 backbones.

We summarize our key observations as follows:

- **Qwen2 on 558K (Table 11)**: The default 1:3 setting achieves the best overall spatial performance. Specifically, it yields the highest results on COCO-QA$_{Spat}$ (62.57), Spatial-MM (41.25), MM-Vet$_{Spat}$ (22.90), and also achieve most tops general metrics. Increasing the stitching ratio beyond this point (e.g., settings (d) and (e)) leads to degradation on both spatial and general benchmarks, indicating over-augmentation.

- **LLaVA on Flickr30K (Table 10)**: The 1:3 setting (**SiTe**$_{default}$) still consistently provides strong performance across tasks. It outperforms other ratios on Spatial-MM (44.97), MME$_{Position}$ (130.70), and general vision-language tasks like COCO-QA (73.74) and MMBench (72.88). Other ratios (settings (1)–(3)) offer marginal or inconsistent gains, and in some cases hurt generalization.

- **Qwen2 on Flickr30K (Table 13)**: In this setting, the 1:3 setting still maintains strong spatial performance, achieving the best results on MME$_{Position}$ (61.00) and balanced results on general tasks. While setting (2) gives slightly better COCO-QA and MMB scores, the default ratio still offers a stable and robust trade-off.

In summary, the 1:3 stitch-to-raw ratio consistently provides a **balanced trade-off between spatial reasoning and general vision-language performance**. Ratios that are too low underutilize the spatial supervision potential of SiTe while overly high ratios risk oversaturating the model with structured spatial prompts, which may harm generalization. Based on these findings, we adopt the 1:3 setting as the default configuration throughout our experiments.

In all experiments, we define the default SiTe configuration as the one where the stitching-to-raw sample ratio is approximately 1:3, which provides a good trade-off performance between spatial understanding and general vision language tasks.

## C.2 More Qualitative Analysis

In this section, we present additional qualitative examples to illustrate both the effectiveness and limitations of the proposed method. **LLaVA** refers to the baseline model. **SiTe**$_{pretrain}$ indicates the model trained with stitched image–caption pairs during the pretraining stage, while **SiTe**$_{SFT}$ refers to the model fine-tuned using structured spatial question–answer pairs.

As shown in Case 1 and Case 2, although all models attend to the relevant objects (e.g., the dog and the people), the baseline model fails to correctly answer the spatial question due to limited understanding of directional language, selecting an incorrect option such as "A. bottom". In contrast, both SiTe-enhanced models correctly interpret the spatial relation and make the right prediction. Case 2 also involves a challenging camera-perspective transformation, which often requires the model to reason from the observer's viewpoint. Here, both SiTe variants successfully leverage spatial knowledge injected through pretraining and fine-tuning, resulting in correct answers.

However, limitations remain in non-camera-perspective scenarios. In Case 3 and Case 4, while all models attend to the appropriate regions (e.g., the glasses in Case 3 and the phone in Case 4), they fail to answer correctly. This suggests that despite improved attention, the models still struggle to generalize spatial understanding across different viewpoints.

These examples highlight both the strengths and boundaries of the proposed method: SiTe improves spatial reasoning under observer-aligned prompts, but further work is needed to enhance generalization under varied spatial perspectives.

## C.3 Standard Deviation of Main Results

We compute the standard deviation of main results in Table 14. As indicated by the standard deviation results, our performance gains are consistent and robust.

**Where is the dogs from the people's perspective? A. bottom B. right C. front D. left**

**Which hand of the boy is the man holding? A. left B. right**

**Which hand is the man using to hold the glasses? A. left B. right**

**In which hand is the man holding the phone? A. left B. right**

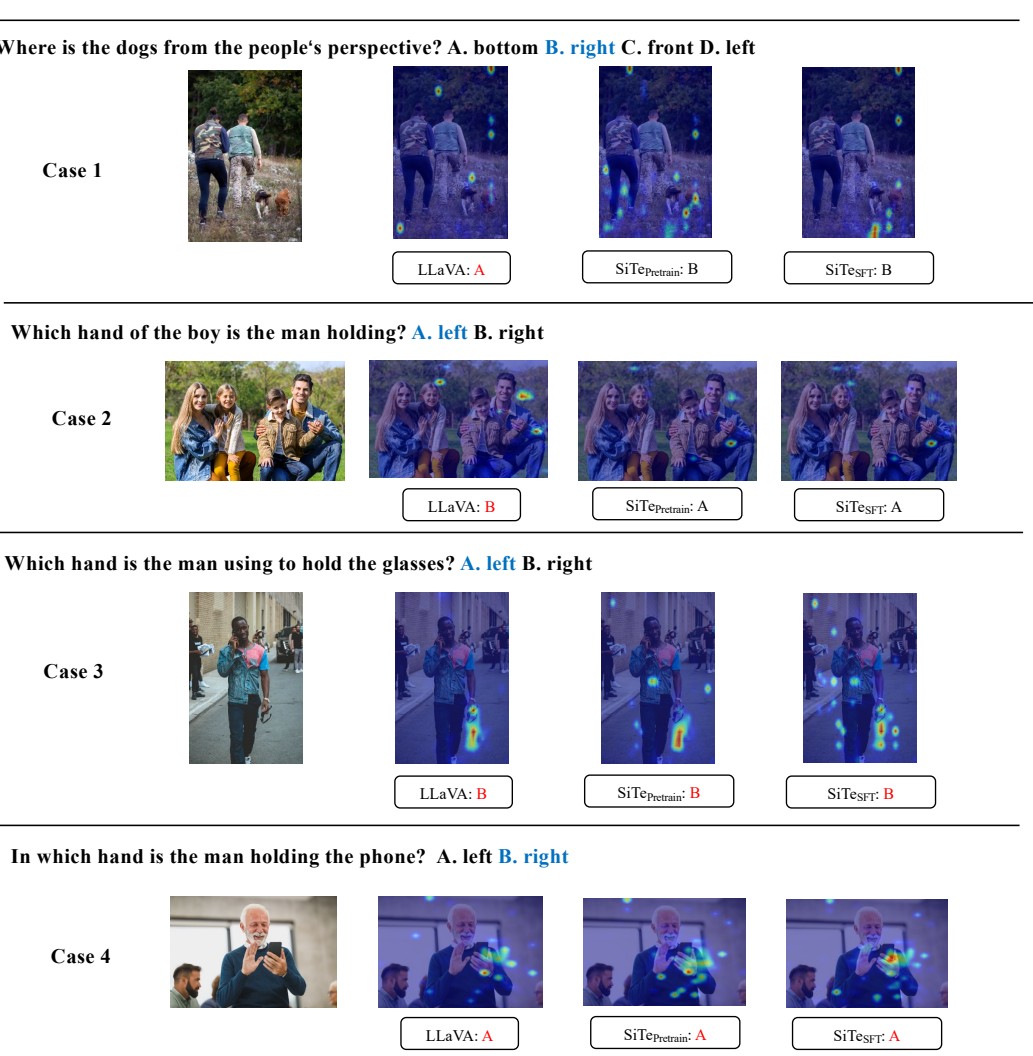

Figure 5: Qualitative comparisons among the baseline LLaVA and our SiTe models.

Table 14: Standard deviation of main results

| Model | Spatial Understanding | | | | General Vision-Language | | | |
|---|---|---|---|---|---|---|---|---|
| | COCO-QA$_{\text{Spat}}$(%) | Spatial-MM(%) | MME$_{\text{Pos}}$ | MM-Vet$_{\text{Spat}}$ | COCO-QA (%) | VQA$_{\text{v2}}$(%) | MMB(%) | MM-Vet |
| *Pretraining Stage* | | | | | | | | |
| LLaVA$_{\text{Baseline}}$ | 67.72±0.10 | 42.02±0.54 | 127.83±1.07 | 26.52±0.08 | 70.52±0.32 | 60.48±0.16 | 73.50±0.14 | 31.11±0.19 |
| LLaVA$_{\text{Rotate}}$ | 68.09±0.17 | 43.01±0.11 | 128.89±1.51 | 29.40±0.03 | 71.36±0.11 | 60.60±0.17 | 74.86±0.05 | 32.20±0.12 |
| LLaVA$_{\text{Crop}}$ | 67.82±0.25 | 42.69±0.24 | 127.78±0.94 | 28.53±0.12 | 70.93±0.18 | 60.37±0.24 | 74.41±0.09 | 31.43±0.25 |
| LLaVA$_{\text{VSR}}$ | 67.85±0.21 | 42.65±0.36 | 121.67±1.01 | 29.42±0.54 | 70.90±0.28 | 60.57±0.25 | 73.88±0.14 | 31.47±0.07 |
| **LLaVA$_{\text{SiTe-rand}}$** | 68.75±0.06 | 43.78±0.14 | 133.33±0.43 | 28.43±0.19 | 71.54±0.27 | 60.53±0.23 | 74.43±0.03 | 31.40±0.04 |
| **LLaVA$_{\text{SiTe-ratio}}$** | 70.06±0.18 | 44.15±0.22 | 132.80±0.97 | 28.68±0.16 | 71.19±0.31 | 60.57±0.20 | 73.64±0.15 | 32.27±0.10 |
| LLaVA$^{\text{flickr}}$ | 68.96±0.21 | 44.03±0.17 | 129.00±0.85 | 25.98±0.14 | 71.75±0.28 | 59.63±0.19 | 72.33±0.13 | 29.54±0.15 |
| LLaVA$^{\text{flickr}}_{\text{Rotate}}$ | 68.03±0.15 | 44.81±0.19 | 129.44±0.78 | 25.86±0.11 | 71.12±0.22 | 59.42±0.18 | 72.36±0.14 | 29.60±0.12 |
| LLaVA$^{\text{flickr}}_{\text{Crop}}$ | 69.34±0.24 | 42.69±0.21 | 131.11±0.92 | 28.73±0.10 | 72.12±0.26 | 60.20±0.21 | 72.68±0.17 | 29.87±0.16 |
| LLaVA$^{\text{flickr}}_{\text{VSR}}$ | 68.26±0.18 | 42.70±0.28 | 128.75±0.97 | 26.18±0.13 | 71.32±0.31 | 59.94±0.24 | 72.87±0.19 | 30.45±0.18 |
| **LLaVA$^{\text{flickr}}_{\text{SiTe-rand}}$** | 71.42±0.13 | 44.97±0.22 | 130.70±0.85 | 26.20±0.14 | 73.74±0.29 | 59.73±0.18 | 72.88±0.14 | 29.59±0.12 |
| **LLaVA$^{\text{flickr}}_{\text{SiTe-ratio}}$** | 71.51±0.15 | 44.31±0.25 | 131.50±0.81 | 28.60±0.12 | 73.89±0.25 | 59.94±0.20 | 71.66±0.16 | 30.97±0.13 |
| Qwen2$_{\text{Baseline}}$ | 62.25±0.16 | 40.85±0.18 | 60.75±0.53 | 20.72±0.10 | 64.72±0.22 | 53.66±0.25 | 61.67±0.19 | 22.48±0.11 |
| Qwen2$_{\text{Rotate}}$ | 60.73±0.14 | 40.78±0.20 | 60.50±0.67 | 19.62±0.15 | 60.22±0.26 | 52.77±0.22 | 66.61±0.23 | 20.97±0.09 |
| Qwen2$_{\text{Crop}}$ | 62.06±0.21 | 40.68±0.15 | 61.25±0.58 | 20.75±0.11 | 64.74±0.19 | 53.28±0.27 | 67.45±0.21 | 22.52±0.14 |
| Qwen2$_{\text{VSR}}$ | 57.18±0.18 | 41.46±0.17 | 58.50±0.73 | 22.40±0.16 | 60.22±0.28 | 54.64±0.23 | 66.58±0.22 | 22.95±0.12 |
| **Qwen2$_{\text{SiTe-rand}}$** | 62.57±0.15 | 41.25±0.22 | 63.00±0.61 | 22.90±0.14 | 65.26±0.23 | 54.18±0.20 | 66.68±0.19 | 22.10±0.13 |
| **Qwen2$_{\text{SiTe-ratio}}$** | 62.52±0.17 | 41.00±0.19 | 68.00±0.88 | 21.00±0.13 | 65.10±0.27 | 54.23±0.21 | 67.57±0.20 | 22.80±0.12 |
| Qwen2$^{\text{flickr}}$ | 47.10±0.20 | 38.90±0.23 | 51.25±0.64 | 10.95±0.10 | 47.90±0.31 | 42.38±0.25 | 50.90±0.19 | 10.50±0.09 |
| Qwen2$^{\text{flickr}}_{\text{Rotate}}$ | 40.08±0.23 | 39.10±0.27 | 58.33±0.75 | 9.50±0.13 | 42.45±0.29 | 39.04±0.28 | 47.86±0.21 | 9.20±0.10 |
| Qwen2$^{\text{flickr}}_{\text{Crop}}$ | 42.37±0.22 | 39.37±0.24 | 64.25±0.78 | 8.30±0.12 | 44.61±0.28 | 40.41±0.25 | 46.43±0.22 | 10.00±0.11 |
| Qwen2$^{\text{flickr}}_{\text{VSR}}$ | 46.00±0.19 | 39.90±0.20 | 67.25±0.70 | 10.15±0.11 | 48.00±0.26 | 40.93±0.24 | 48.71±0.20 | 9.60±0.09 |
| **Qwen2$^{\text{flickr}}_{\text{SiTe-rand}}$** | 47.71±0.17 | 39.35±0.18 | 61.67±0.82 | 12.47±0.14 | 48.68±0.25 | 42.95±0.21 | 51.33±0.18 | 10.80±0.11 |
| **Qwen2$^{\text{flickr}}_{\text{SiTe-ratio}}$** | 49.04±0.18 | 40.04±0.20 | 56.00±0.69 | 13.10±0.15 | 47.38±0.27 | 43.91±0.23 | 51.31±0.17 | 10.60±0.10 |
| *Supervised Fine-tuning Stage* | | | | | | | | |
| **LLaVA$^{\text{1K}}_{\text{SiTe-rand}}$** | 68.81±0.26 | 43.16±0.27 | 128.70±0.08 | 27.06±0.11 | 71.74±0.47 | 61.17±0.29 | 74.60±0.26 | 31.20±0.03 |
| **LLaVA$^{\text{1K}}_{\text{SiTe-ratio}}$** | 68.35±0.21 | 46.96±0.23 | 136.00±0.92 | 29.70±0.14 | 71.32±0.33 | 60.92±0.27 | 73.37±0.19 | 31.54±0.10 |
| **LLaVA$^{\text{5K}}_{\text{SiTe-rand}}$** | 67.75±0.04 | 46.21±0.14 | 139.26±0.31 | 28.10±0.23 | 70.96±0.13 | 60.38±0.20 | 74.53±0.18 | 30.69±0.24 |
| **LLaVA$^{\text{5K}}_{\text{SiTe-ratio}}$** | 68.37±0.22 | 48.58±0.26 | 141.00±1.03 | 31.05±0.16 | 70.92±0.28 | 60.82±0.25 | 73.76±0.20 | 32.15±0.12 |
| **Qwen2$^{\text{1K}}_{\text{SiTe-rand}}$** | 58.53±0.47 | 41.24±0.03 | 65.33±0.54 | 23.10±0.44 | 61.77±1.32 | 54.58±0.03 | 66.37±0.11 | 23.93±0.22 |
| **Qwen2$^{\text{1K}}_{\text{SiTe-ratio}}$** | 59.66±0.32 | 42.31±0.27 | 62.75±0.63 | 24.07±0.21 | 62.59±0.35 | 55.30±0.25 | 65.75±0.19 | 22.80±0.18 |
| **Qwen2$^{\text{5K}}_{\text{SiTe-rand}}$** | 59.95±0.53 | 42.22±0.21 | 61.67±0.53 | 23.20±0.37 | 63.00±1.10 | 54.75±0.14 | 66.43±0.13 | 23.93±0.19 |
| **Qwen2$^{\text{5K}}_{\text{SiTe-ratio}}$** | 60.22±0.35 | 41.56±0.29 | 64.50±0.59 | 25.10±0.22 | 62.86±0.41 | 54.77±0.20 | 67.00±0.18 | 24.27±0.15 |
| HALVA$^{*}_{\text{Baseline}}$ | 63.16±0.14 | 43.07±0.26 | 135.00±0.94 | 25.70±0.13 | 67.12±0.25 | 61.67±0.18 | 72.44±0.16 | 30.00±0.11 |
| **HALVA$_{\text{SiTe}}$** | 64.77±0.18 | 44.15±0.22 | 123.33±0.83 | 26.10±0.12 | 68.54±0.27 | 61.03±0.20 | 71.54±0.15 | 30.80±0.10 |

