# OpenReview forum: "Stitch and Tell: A Structured Data Augmentation Method for Spatial Understanding"
_NeurIPS.cc/2025/Conference — NeurIPS 2025 poster_

### Official Review · Reviewer_3Q8z · 2025-06-29

**Clarity:** 4
**Significance:** 3
**Originality:** 3
**Rating:** 5
**Confidence:** 4

**Summary:**

This paper introduces "Stitch and Tell" (`SiTe` in short), a simple data augmentation strategy for multimodal data that aims at improving spatial understanding of Large Multimodal Models (LMMs).

SiTe works in two steps:
1. It stitches two randomly sampled images onto the vertical or horizontal axis (randomly selected);
2. It constructs a spatially-aware caption following a template (for example: "To the left is {caption1}, while the right part contains {caption2}").

SiTe also has a version for Instruction-Tuning data, where noun phrases are extracted from the texts and templates constitute spatially-aware questions, such as "Is {object in left image} to the left of {object in right image}"?

In this work, the first variant is used as a data augmentation strategy for pretraining (on image-caption pairs from llava-595k), while the second variant is used for instruction tuning (llava-665k), and evaluation is conducted on both (i) benchmarks probing spatial understanding and (ii) general multimodal benchmarks.

The paper experiments with LLaVA-v1.5-7B, LLaVA-Qwen2-7B and HALVA, showing that SiTe is effective for both Pretraining (PT) and SFT while not harming performance on standard multimodal benchmarks (e.g., MMBench).

**Questions:**

For a productive rebuttal, I would discuss the weaknesses above as well as the following questions:

1. A by-product of SiTe is that the average image resolution processed by the LMM is higher. Did the authors evaluate if, as a side effect, also high-resolution image understanding of LMMs trained with SiTe data is improved? I think this would be a nice finding if true;
2. The image stitching algorithm (Algorithm 1) states that images are pasted on a blank canvas, which is sized according to the larger vertical / horizontal edge between the two sampled images. This inevitably leads to areas of the canvas which will remain blank, which are possibly very large if the sampled images have large differences in size, and consequently entails throughput inefficiencies since there will be many unneeded visual tokens. Naive solutions to this would be proportional resizing of the image with the smaller edge, or sampling / grouping images of similar size. Did the authors consider these aspects?
3. The experimental section says that 5 runs were executed. Could the authors please report the standard deviation as well? I think this would be important to understand if the improvements provided by SiTe are significant.

**Ethical Concerns:**

["NO or VERY MINOR ethics concerns only"]

**Final Justification:**

The discussion with the authors was positive, introducing results for 3D spatial understanding, exploring more efficient ways to stitch images together, and examining high-resolution image understanding as a by-product of the proposed method. My score was positive from the beginning, and I am keeping it.

**Limitations:**

The paper contains a limitations section in the appendix (which mentions 3D spatial understanding). I think this is sufficient, even though ideally it should fit the main body of the paper.

**Paper Formatting Concerns:**

I don't see any formatting concerns.

**Quality:**

3

**Strengths And Weaknesses:**

**Strengths.**
1. This paper introduces a simple idea, which I believe we should promote and not penalize;
2. Spatial Understanding is overall an important problem to tackle, as it serves many applications (*e.g.*, robotics or autonomous driving);
3. SiTe is presented in two variants (PT and SFT), showing effectiveness in both scenarios;
4. SiTe is used to augment only a subset of PT or SFT data, showing that spatial understanding can be boosted without compromising general multimodal knowledge;
5. By reducing the overall number of data points fairly, SiTe also incurs a lower computational budget than standard SFT since the overall number of data points is reduced (stitched images are not re-used data points);


**Weaknesses.**
1. The related work of this paper introduces other multimodal data-augmentation strategies. I understand some of these may be heavier compared to SiTe, but I wonder why none of them are empirically validated in this work (even though they are not "spatially-aware", showing they don't translate into any downstream gain for spatial understanding might make the paper more complete);
2. Given the nature of flat image stitching (e.g., images are stitched on a flat 2D canvas), I wonder whether SiTe hurts or benefits spatial understanding when it comes to a notion of depth (so, spatial relationships as "{obj1} is behind of {obj2}"). I think it would be nice to also evaluate on CV-Bench [a] (both the 2D and 3D subsets), which has gained quite some popularity in the multimodal community over the past year. The 3D subset would specifically serve as a tool to probe if SiTe would unlock "depth-aware" spatial understanding as a by-product.

**References.**
[a] Tong, Peter, et al. "Cambrian-1: A fully open, vision-centric exploration of multimodal llms." NeurIPS 2024.

---

> ### Author Rebuttal · Authors · 2025-07-31
>
> **Dear Reviewer 3Q8z, we truly appreciate your thorough review, positive feedback, and the time and effort you devoted to strengthening our paper. We are grateful that you highlighted the simplicity and practical value of our approach, as well as the importance of spatial understanding for real-world applications. We are also encouraged by your recognition of the effectiveness of SiTe. Your thoughtful remarks are highly motivating for us.**
>
> ---
> > **`Q1:Did the authors evaluate if, as a side effect, also high-resolution image understanding of LMMs trained with SiTe data is improved?`**
>
> R1: Thanks and we really like this insightful comment! We conducted additional experiments on the HR-Bench 8K [a] and V-Star [b] high-resolution datasets. **As shown in the table below, models enhanced with SiTe consistently achieve better performance on these high-resolution benchmarks.** This result motivates us to further explore the potential of SiTe for more aspects in visual understanding task!
>
> |Model|HR-Bench|V*|
> |----------------|---------|------|
> |$\mathbf{LLaVA_{Baseline}}$|30.90|48.34|
> |$\mathbf{LLaVA_{SiTe}}$|**32.56**|**49.95**|
> |$\mathbf{Qwen_{Baseline}}$|32.25|37.83|
> |$\mathbf{Qwen_{SiTe}}$|**32.67**|**38.92**|
>
> [a] Wang, Wenbin, et al. "Divide, conquer  and combine: A training-free framework for high-resolution image  perception in multimodal large language models." Proceedings of the AAAI Conference on Artificial Intelligence. Vol. 39. No. 8. 2025.
>
> [b] Wu, Penghao, and Saining Xie. "V*: Guided visual search as a core mechanism in multimodal llms." Proceedings of the IEEE/CVF Conference on Computer Vision and Pattern Recognition. 2024.
>
>
> ---
> > **`Q2: Did the authors consider the resizing or grouping the images of similar size?`**
>
> R2：Thank you for your thoughtful suggestion! We found your idea very interesting and have incorporated it by improving the SiTe stitching strategy.
>
> $\mathbf{SiTe_{resize}}$: For left-right stitching, we resize the shorter image so that both images have the same height; for top-down stitching, we resize the narrower image so that both images have the same width.
>
> $\mathbf{SiTe_{ratio}}$: We first categorize images based on their aspect ratios. Images with width/height > 1.2 are grouped for top-down stitching, and images with height/width > 1.2 are grouped for left-right stitching.
>
> $\mathbf{SiTe_{default}}$ refers to the default random stitching method described in the paper.
>
> We conducted experiments using these variants in the pretraining stage of LLaVA with the blip_laion_cc_sbu_558K(558K) dataset. The results are shown below:
>
> |Model|COCO-QA-spatial|Spatial-MM|MME-pos|MM-vet-spat|COCO-QA|MMB|VQA_v2|MM-Vet-total|
> |---|---|---|---|---|----|---|---|---|
> |$\mathbf{Baseline}$|67.72|42.02|127.83|26.52|70.52|73.50|60.48|31.11|
> |$\mathbf{Site_{resize}}$|68.19|42.98|130.25|27.97|70.94|73.24|60.52|31.46|
> |$\mathbf{Site_{ratio}}$|**70.26**|**44.15**|132.80|**28.68**|71.19|73.64|**60.57**|**32.27**|
> |$\mathbf{Site_{default}}$|68.75|43.78|**133.33**|28.43|**71.54**|**74.43**|60.53|31.40|
>
> As shown, $\mathbf{SiTe_{ratio}}$ achieves better on most benchmarks. We suppose this is because grouping images by ratio ensures that the stitched images have more realistic proportions, avoiding unnatural shapes that are too wide or too tall. $\mathbf{SiTe_{resize}}$ may create stitched images with resolutions that differ from the testset, which could negatively affect performance.
>
> ---
> > **`Q3: Did SiTe hurt or benefits spatial understanding when it comes to a notion of depth?`**
>
> R3：Thank you for raising this concern! We addressed it by further evaluating our models on CV-Bench[a] and on RealWorldQA[b], which focuses on basic real-world spatial understanding in multimodal models. The results are presented below.
> |Model|CV-Bench 2d|CV-Bench 3d|RealWorldQA|
> |---|---|---|---|
> |$\mathbf{LLaVA_{Baseline}}$|52.56|33.43|55.12|
> |$\mathbf{LLaVA_{SiTe}}$|**54.28**|**35.50**|**55.21**|
> |$\mathbf{Qwen_{Baseline}}$ |46.77|47.63|52.28|
> |$\mathbf{Qwen_{SiTe}}$ |**48.16**|**50.36**|**54.44**|
>
> As shown, **the injection of spatial supervision via SiTe leads to improved model performance on out-of-domain spatial relation tasks and demonstrates strong generalization to more complex spatial reasoning scenarios in CV-Bench and RealWorldQA**. A reasonable hypothesis is that the model benefits from its improved understanding of fundamental directional relationships.
>
> [a] Tong, Peter, et al. "Cambrian-1: A fully open, vision-centric exploration of multimodal llms." NeurIPS 2024.
>
> [b] X.AI. Realworldqa. Blog post, July 2025. Accessed: 2025-07-30.
>
> ---
> > **`Q4：Could the authors please report the standard deviation as well?`**
>
> R4: Of course! We compute the standard deviation in the below table. Due to space limitations, we show a subset of the results here.
>
> |Model|COCO-QA_Spat|Spatial-MM|MME_Pos|MM-Vet_Spat|COCO-QA|VQAv2|MMB|MM-Vet|
> |---|---|---|---|---|---|---|---|---|
> |$\mathbf{LLaVA_{Baseline}}$|67.72±0.10|42.02±0.54|127.83±1.07|26.52±0.08|70.52±0.32|60.48±0.16|73.50±0.14|31.11±0.19|
> |$\mathbf{LLaVA_{Rotate}}$|68.09±0.17|43.01±0.11|128.89±1.51|29.40±0.03|71.36±0.11|60.60±0.17|74.86±0.05|32.20±0.12|
> |$\mathbf{LLaVA_{Crop}}$|67.82±0.25|42.69±0.24|127.78±0.94|28.53±0.12|70.93±0.18|60.37±0.24|74.41±0.09|31.43±0.25|
> |$\mathbf{LLaVA_{VSR}}$|67.85±0.21|42.65±0.36|121.67±1.01|29.42±0.54|70.90±0.28|60.57±0.25|73.88±0.14|31.47±0.07|
> |$\mathbf{LLaVA_{SiTe}}$|68.75±0.06|43.78±0.14|133.33±0.43|28.43±0.19|71.54±0.27|60.53±0.23|74.43±0.03|31.40±0.04|
> |$\mathbf{LLaVA_{SiTe}^{1K}}$|68.81±0.26|43.16±0.27|128.70±0.08|27.06±0.11|71.74±0.47|61.17±0.29|74.60±0.26|31.20±0.03|
> |$\mathbf{LLaVA_{SiTe}^{5K}}$|67.75±0.04|46.21±0.14|139.26±0.31|28.10±0.23|70.96±0.13|60.38±0.20|74.53±0.18|30.69±0.24|
> |$\mathbf{Qwen2_{SiTe}^{1K}}$|58.53±0.47|41.24±0.03|65.33±0.54|23.10±0.44|61.77±1.32|54.58±0.03|66.37±0.11|23.93±0.22|
> |$\mathbf{Qwen2_{SiTe}^{5K}}$|59.95±0.53|42.22±0.21|61.67±0.53|23.20±0.37|63.00±1.10|54.75±0.14|66.43±0.13|23.93±0.19|
>
> **As indicated by the standard deviation results, the performance gains are consistent and robust**, and we will add the whole results in the revised paper.
>
> ---
> > **`Q5：Are you considering comparing other data augmentation methods, even they are more heavier?`**
>
> R5：Thanks for the suggestion! Following approaches similar to [a] and [b], we performed data augmentation by using a multimodal large language model to rewrite the original captions and inject spatial information into the training data.
> We used the following prompt for GPT-4o to rewrite captions based on the 558K.
> >Based on the image and its original caption, Your task is to generate a new caption that prioritizes spatial relationships, no more than 20 words.
>
> To evaluate the rewritten captions[c], we used Qwen2.5VL-7B as a judge with the following prompt:
> >Your task is to compare two captions for an image and determine which better meets the evaluation criteria. (Print ONLY 1, -1, or 0 as the result)
> >Evaluation Criteria:
> >1. Accuracy: How accurately the caption describes the actual content of the image.
> >2. Positional Details: The quantity and quality of positional/directional descriptions.
> >Result: Return 1 if Caption1 is better, -1 if Caption2 is better, and 0 if equally good.
> >Caption1: {new_caption}
> >Caption2: {old_caption}
>
> The results are as follows:
> ```
> Score=1: 94.36%, Score=0: 2.97%, Score=-1: 2.68%
> ```
>
> These results confirm that the rewritten captions significantly improve spatial semantic richness and descriptive accuracy.
>
> We then used the rewritten captions to train LLaVA in the pretraining stage. The results are shown below:
>
> |Model|COCO-QA-spatial|Spatial-MM|MME-pos|MM-vet-spat|COCO-QA|MMB|VQA_v2|MM-Vet-total|
> |----|---|---|---|---|---|---|---|---|
> |$\mathbf{LLaVA_{Baseline}}$|67.72|42.02|127.83|26.52|70.52|73.50|60.48|31.11|
> |$\mathbf{LLaVA_{Rewrite}}$|**70.39**|**44.95**|129.50|**30.40**|**73.01**|73.34|**61.92**|**32.15**|
> |$\mathbf{LLaVA_{SiTe}}$|68.75|43.78|**133.33**|28.43|71.54|**74.43**|60.53|31.40|
>
> As shown, rewriting captions leads to significant performance improvements, demonstrating the importance of explicitly insert spatial information in captions. This approach also produces captions that are more coherent and natural, injecting richer supervisory signals and more detailed information.
>
> However, it is worth noting that caption rewriting for the 558K dataset cost approximately $560 using the GPT-4o API, and this expense would increase significantly for even larger datasets (not including the additional cost of caption quality evaluation).
>
> As described in our paper, SiTe focuses on providing a lightweight, effective, and unambiguous way to inject spatial supervision. In future work, we plan to explore combining SiTe with caption rewriting to further enhance spatial semantic richness in the training data.
>
> [a] Chen G H, et al. Allava: Harnessing gpt4v-synthesized data for lite vision-language models[J]. arXiv preprint arXiv:2402.11684, 2024.
>
> [b] Chen L, et al. Sharegpt4v: Improving large multi-modal models with better captions[C]//European Conference on Computer Vision. Cham: Springer Nature Switzerland, 2024: 370-387.
>
> [c] Lee, Yebin, et al. Fleur: An explainable reference-free evaluation metric for image  captioning using a large multimodal model. arXiv preprint arXiv:2406.06004 (2024).
>
> ---
> We thank the review again for this **insightful** and **interesting** comment. We will include the above analysis in the revised paper to make the paper more complete. Thanks again!

---

> > ### Comment · Reviewer_3Q8z · 2025-08-02
> >
> > Dear Authors,
> >
> > Thank you for the detailed rebuttal. Let me reply to each point:
> >
> > **High-res understanding as a by-product of SiTe.** It's nice to see such a side-effect. If you have sufficient compute, I'd suggest reporting the complete set of HR experiments on a significant subset of the models you pre-trained or fine-tuned in the revised version of the paper.
> >
> > **Strategies to mitigate blank visual tokens.** Thank you for running these experiments! I think how you performed grouping based on aspect ratios makes sense, and it's nice to see it improves over a vanilla implementation of SiTe. There are probably more token-efficient ways to do so (e.g., once the left image is sampled, one might greedily pick the right image as the one leading to the *least* number of blank pixels or tokens), either in-batch or offline as you've done here. In any case, similarly to above, I'd suggest reporting the improved variant in the revised manuscript if you have sufficient compute, and you find consistent improvement over experimental setups.
> >
> > About all other points, I think your replies are sufficient, and I do not have further comments. Please make sure the corresponding changes, pointers, and clarifications are added to the revised paper.
> >
> > All the best,\
> > Reviewer `3Q8z`

---

> > > ### Author Response · Authors · 2025-08-03
> > >
> > > Thank you very much for your positive recognition and constructive feedback. We are glad that our response addressed your concerns.
> > >
> > > In the revised paper, we will include a more comprehensive set of experiments on High-res datasets, both in the pretraining and supervised fine-tuning stages.
> > >
> > > In addition, we will explore more efficient pairing strategies to further align the distribution of stitched images with the test sets and to generate more coherent stitched captions.
> > >
> > > We appreciate your suggestions and will make sure to clarify these points in the revised version. Thank you again for your detailed review and valuable guidance!

---

### Official Review · Reviewer_htxQ · 2025-07-01

**Clarity:** 3
**Significance:** 2
**Originality:** 3
**Rating:** 3
**Confidence:** 4

**Summary:**

The paper proposes SiTe, a data augmentation method to enhance the spatial understanding capabilities of MLLMs. The approach consists of two key steps: (1) stitching two images along a spatial axis to create a combined image, and (2) generating spatially grounded question-answer pairs based on the new layout. Experiments on baseline models demonstrate improvements in spatial perception tasks.

**Questions:**

- How many total samples were used for training. Do you use 1K/5K augmented data along with original data?
- What is the distribution of different stitched image types and question types?
- Would the method benefit from testing on SOTA MLLMs?
- Is the proposed method able to handle more complex reasoning about occlusion, depth, and partial visibility?

**Ethical Concerns:**

["NO or VERY MINOR ethics concerns only"]

**Final Justification:**

After the rebuttal, I remain concerned about the inherent limitations of SiTe, a binary ("yes" or "no") answer format, which may restrict its generalization to more complex spatial reasoning tasks. This simplicity could explain the observed performance drop in general vision-language tasks, as also noted by other reviewers. After carefully considering the rebuttal and other reviews, I still find the practicality of SiTe augmentation limited in its current form. Given these unresolved issues, I would like to maintain my original score as borderline reject.

**Limitations:**

Yes

**Paper Formatting Concerns:**

No major issues found.

**Quality:**

2

**Strengths And Weaknesses:**

Strengths
- The proposed method is simple yet effective, requiring minimal modifications to existing pipelines.
- SiTe can augment existing training datasets without costly manual annotations, making it scalable.

Weaknesses
- The paper lacks clarity on dataset details, such as total training size, composition, and diversity.
- Experiments are conducted on baseline models including LLaVA and HALVA. Validation on state-of-the-art MLLMs (e.g., Qwen2-VL, Intern-VL) is missing. It’s unclear if the proposed method can achieve consistent improvement on these stronger models.
- The proposed stitching paradigm generates highly synthetic images (e.g., rigid left-right concatenation) that may not reflect the complexity of natural spatial relationships. Real-world scenes often involve occlusion and are influenced by depth and 3D layout. Therefore, while the method improves performance on simplified spatial QA tasks, its generalization to real-world scenarios is limited since spatial understanding requires inferring latent relationships (e.g.,  to answer the question "Is the cup behind the book?").

---

> ### Author Rebuttal · Authors · 2025-07-31
>
> **Dear Reviewer htxQ, we thank for your careful and helpful comments on our work. We appreciate the reviewer’s recognition of the simplicity and effectiveness of our approach, as well as the scalability benefits of SiTe. Below are our responses to all the concerns you raised, which will be incorporated into the updated version to further enhance the quality of our submission.**
>
> ---
> > **`Q1: How many total samples were used for training? Do you use 1K/5K augmented data along with original data? What is the total training size, composition, and diversity?`**
>
> R1：Thank you for your question! SiTe applies data augmentation in both the pretraining and fine-tuning stages.
>
> In the pretraining stage, we experimented with two datasets. The first is blip_laion_cc_sbu_558K: $\mathbf{LLaVA_{Baseline}}$ was trained on 558K image–caption pairs, while $\mathbf{LLaVA_{SiTe}}$ used 458K pairs, since one-third of the data was stitched and duplicate pairs were removed (as shown in **Table 3** of the manuscript). The second dataset is flickr-30K: $\mathbf{LLaVA^{Flickr}}$ was trained on 30K image–caption pairs, while $\mathbf{LLaVA_{SiTe}^{Flickr}}$ used 24K pairs, with one-third of the data being stitched and deduplicated.
>
> The number of data in pretraining stage is shown in the figure below (taking LLaVA as an example):
> | | $\mathbf{LLaVA_{Baseline}}$ | $\mathbf{LLaVA_{SiTe}}$ | $\mathbf{LLaVA^{Flickr}}$ | $\mathbf{LLaVA_{SiTe}^{Flickr}}$ |
> |---|----|----|----|-----|
> | **Num of Pretrain Data** | 558K  | 458K (2/3 Raw + 1/3 SiTe)| 30K | 24K (2/3 Raw + 1/3 SiTe) |
> | **Num of SFT Data**  | 665K  | 665K | 665K   | 665K  |
>
> In the supervised fine-tuning stage, we performed partial stitching on the original llava_v1_5_mix665k dataset. Specifically, we randomly sampled 1K or 5K augmented samples and added them to the original fine-tuning dataset llava_v1_5_mix665k(665K) for full-parameter fine-tuning. This design helps maintain model generalization and avoid overfitting.
>
> The number of data in supervised fine-tuning stage is shown in the figure below :
>
> | | $\mathbf{LLaVA_{SiTe}^{1K}}$ | $\mathbf{LLaVA_{SiTe}^{5K}}$ |
> |-------|-----|-----|
> | **Pretrain Data**| 558K  | 558K  |
> | **SFT Data** | 665K + 1K (SiTe)  | 665K + 5K (SiTe) |
>
> Regarding template diversity, SiTe employs a variety of directional stitching templates, as detailed in Appendix Tables 5, 6, 7, and 8. During augmentation, the stitching format is randomly sampled from these templates, so each template is selected with equal probability. As for the diversity of the training set, since SiTe does not introduce any additional supervision beyond the templates, **the diversity of the augmented data remains consistent with that of the original dataset.**
>
> ---
> > **`Q2: What is the distribution of different stitched image types and question types?`**
>
> R2：Thank you for your question. As stated in the paper, the proportion of horizontally and vertically stitched images, as well as question types, is set to 1:1 by design:
>
> *We apply both horizontal and vertical stitching in a 1:1 ratio.*
>
> And we specify the default SiTe setting:
>
> *By default, the stitched data make up one-third of the total set.*
>
> These details can be found in Section 4.2 (Pretraining Stage) of our paper.
>
> ---
> > **`Q3: Would the method benefit from testing on SOTA MLLMs?`**
>
> R3：Thank you for your suggestion! Given the architectural similarities between QwenVL and InternVL, and considering that we have already included experiments with LLaVA-Qwen2 in the main text, we further conducted preliminary experiments on InternVL2-4B. We modified the training process to follow a Pretrain + SFT paradigm and applied SiTe augmentation during the pretraining stage. Specifically, we used both the blip_laion_cc_sbu_558K dataset and its SiTe-augmented version (using the default setting).
>
> |Model|COCO-QA-spatial|Spatial-mm|MME-pos|MM-vet-spat|COCO-QA|MMB|VQA_v2|MM-Vet-total|
> |---|----|---|---|---|---|---|---|---|
> |$\mathbf{InternVL2_{Baseline}}$|51.01%|40.51%|**136.00**|24.1%|55.69%|66.57%|55.10%|24.4%|
> |$\mathbf{InternVL2_{SiTe}}$|**52.03%**|**41.75%**|129.00|**29.3%**|**56.25%**|**68.35%**|**55.30%**|**25.6%**|
>
> As shown, InternVL-SiTe consistently outperforms the baseline InternVL on most spatial understanding tasks, while also maintaining strong generalization on general benchmarks. **This demonstrates that SiTe can also benefit SOTA MLLMs.** These results will be incorporated into the updated version to further enhance the quality of our submission.
>
> ---
> > **`Q4: Is the proposed method able to handle more complex reasoning about occlusion, depth, and partial visibility?`**
>
> R4: Thank you for your insightful comment! We acknowledge that the current stitching paradigm does not fully capture the complexity of natural spatial relationships. However, SiTe is designed to provide a lightweight and effective way to inject explicit spatial supervision at minimal cost, without requiring complex data collection or annotation.
>
> To evaluate SiTe's generalization, we selected out-of-distribution spatial relation samples from COCOQA (COCOQA-ood), which include spatial relation keywords such as "behind", "in front of", "close to", "far away from", "inside", "outside", "within", "internal", "external", "ahead of", and "adjacent to". We further evaluated our models on CV-Bench[a], which specifically tests complex spatial reasoning, including depth and occlusion, and on RealWorldQA[b], which is designed to evaluate basic real-world spatial understanding capabilities of multimodal models. The results are shown in the table below:
>
> | Model   | COCOQA-ood | CV-Bench 2d | CV-Bench 3d |RealWorldQA|
> |-----------------|------------|-------------|-------------|---|
> | $\mathbf{LLaVA_{Baseline}}$  | 70.38%  | 52.56% | 33.43%|55.12%|
> | $\mathbf{LLaVA_{SiTe}}$  | **71.47%** | **54.28%**  | **35.50%**  |**55.21%**|
> | $\mathbf{Qwen_{Baseline}}$ | 69.24%  | 46.77%  | 47.63%  | 52.28%|
> | $\mathbf{Qwen_{SiTe}}$ | **71.34%** | **48.16%**  | **50.36%**  |**54.44%**|
>
> As shown, the injection of spatial supervision via SiTe leads to improved model performance on out-of-domain spatial relation tasks and demonstrates strong generalization to more complex spatial reasoning scenarios in CV-Bench and RealWorldQA. A reasonable hypothesis is that the model benefits from its improved understanding of fundamental directional relationships. We agree that more realistic and comprehensive spatial augmentation is an important direction for future work.
>
> **In summary, although SiTe does not directly address occlusion or 3D reasoning, our results show that it improves generalization to these more complex spatial tasks.**
>
> [a] Tong, Peter, et al. "Cambrian-1: A fully open, vision-centric exploration of multimodal llms." NeurIPS 2024.
>
> [b] X.AI. Realworldqa. Blog post, July 2025. Accessed: 2025-07-30
>
> ---
> We thank the reviewer again for these careful comments and hope these responses address your concerns!

---

> > ### Comment · Reviewer_htxQ · 2025-08-04
> >
> > Thank you for your detailed rebuttal and additional experimental results. However, I still have a remaining question regarding the scale of SiTe-augmented data used in the SFT stage.
> >
> > In your experiments, $\mathbf{LLaVA_{SiTe}^{1K}}$ and $\mathbf{LLaVA_{SiTe}^{5K}}$ introduce only 1K/5K SiTe-augmented samples during SFT, which is significantly less than the pretraining data augmentation in $\mathbf{LLaVA_{SiTe}}$ (where 1/3 of the pretraining data is augmented with SiTe). Could you clarify why such a small scale of SiTe data was chosen for SFT?
> >
> > Moreover, the experimental results show that $\mathbf{LLaVA_{SiTe}^{1K}}$ does not exhibit a clear advantage over LLaVA-SiTe, and expanding to 5K samples $\mathbf{LLaVA_{SiTe}^{5K}}$ even leads to performance degradation on several benchmarks—falling below the original LLaVA baseline. Does this suggest that SiTe is only effective during pretraining and has limited utility in the SFT stage?

---

> > > ### Author Response · Authors · 2025-08-04
> > >
> > > Thank you for your questions!  Let us reply to each point:
> > >
> > > 1. **The scale of SiTe-augmented data used in the SFT stage**:
> > >
> > > The supervised fine-tuning (SFT) stage is intended for targeted task adaptation rather than for large-scale knowledge injection. **Our goal was to examine whether a small amount of spatially augmented data could improve spatial reasoning while maintaining strong performance on general benchmarks**. Our results show that even with only 1K or 5K SiTe-augmented samples, we observe clear improvements on spatial tasks, demonstrating the efficiency and effectiveness of SiTe in the SFT stage. In contrast, the pretraining stage aims to broadly enhance the model’s image-text alignment abilities, which typically requires a much larger scale of data.
> > >
> > > 2. **The impact of scaling up SiTe data during SFT**:
> > >
> > > It is common in multimodal learning that introducing a large proportion of domain-specific data during fine-tuning can lead to overfitting to that domain and a corresponding drop in generalization performance on more diverse or general tasks. This is consistent with our observation that increasing SiTe-augmented data to 5K in SFT brings diminishing or even negative returns on some general benchmarks.
> > >
> > > Moreover, it is important to note that the SiTe-augmented data (665K) used in SFT is not tailored to the precise evaluation domains of the downstream benchmarks. **The fact that we still observe improvements in most cases further validates the robustness and generalization of SiTe augmentation**.
> > >
> > > In summary, SiTe is effective in both the pretraining and SFT stages, though the optimal scale of augmentation depends on the specific goals and trade-offs of each stage.
> > >
> > > ---
> > >
> > > We hope this explanation resolves your concerns. Please feel free to let us know if you have any additional questions, and we will be happy to provide additional clarification :)

---

> > > > ### Comment · Reviewer_htxQ · 2025-08-06
> > > >
> > > > Thank you for your follow-up responses and your clarification.
> > > >
> > > > I acknowledge your point about domain-specific data potentially harming generalization. However, I remain concerned because the amount of SiTe-augmented data used (1K/5K out of 665K) is too small to be considered "a large proportion."
> > > >
> > > > My primary concern is the method's poor scalability during the SFT stage. Your results indicate that increasing from 1K to 5K SiTe samples yields minimal improvements even on spatial-related benchmarks (COCO-QA_spat and MM-Vet_spat) compared to both the SiTe pretraining baseline and the 1K SiTe SFT baseline.However, it actively degrades performance on general benchmarks.
> > > >
> > > > This raises questions about the method’s scalability and practical applicability. If such a small increase in augmented data (less than 1% of total training data) can hurt general model capabilities while providing negligible gains on target tasks, this suggests potential limitations in the approach's robustness. Given that current MLLMs rely heavily on substantial amounts of training data, this sensitivity to small-scale augmentation could limit its real-world deployment.

---

> > > > > ### Author Response · Authors · 2025-08-06
> > > > >
> > > > > Thank you for your thoughtful concern.
> > > > >
> > > > > Our QA templates are intentionally constructed **independently** of the evaluation benchmarks and mainly use a single-turn, binary (“yes” or “no”) answer format. To further clarify the impact, we analyzed the answer distribution in the original 665K SFT training set: only 795 examples are labeled “no” and 611 as “yes” within single-turn dialog responses. By adding several thousand SiTe-augmented QA pairs using this format, the relative frequency of such responses in training data increases substantially. This shift in answer pattern can disproportionately affect the model’s response distribution, especially in general benchmarks with more diverse answer formats. This helps explain why even small-scale SiTe augmentation may noticeably impact generalization.
> > > > >
> > > > > However, the real strength of SiTe lies in its flexibility. For scenarios that require both strong general capabilities and enhanced spatial reasoning, applying SiTe at the pretraining stage is most effective, allowing broad spatial supervision without harming generalization. If a downstream application prioritizes spatial reasoning, SiTe templates during **SFT can be adapted to better match the evaluation format, further improving performance in that setting**.
> > > > >
> > > > > **In summary, SiTe’s flexibility across training stages allows users to tailor their augmentation strategy—whether to boost domain-specific reasoning or maintain broad multimodal competence.** This adaptability is a core strength of SiTe and supports its application in a variety of real-world tasks.
> > > > >
> > > > > ---
> > > > > We hope the above explanation can address your concern! :)

---

> > > > > > ### Comment · Reviewer_htxQ · 2025-08-07
> > > > > >
> > > > > > Thank you for the detailed clarifications. While I appreciate the discussion on SiTe augmentation, I remain concerned about the inherent limitations of its binary ("yes" or "no") answer format, which may restrict its generalization to more complex spatial reasoning tasks. This simplicity could explain the observed performance drop in general vision-language tasks, as also noted by other reviewers. After carefully considering the rebuttal and other reviews, I still find the practicality of SiTe augmentation limited in its current form. Given these unresolved issues, I would like to maintain my original score as borderline reject.

---

> > > > > > > ### Author Response · Authors · 2025-08-07
> > > > > > >
> > > > > > > Thank you for your continued feedback and we provide our responses below:
> > > > > > >
> > > > > > > First, While our current experiments use binary (“yes”/“no”) QA templates for simplicity, SiTe is inherently extensible and not limited to this format. The augmentation pipeline readily supports arbitrary and customizable templates,
> > > > > > >
> > > > > > > - **Yes/No**
> > > > > > > - **Multiple-choice**
> > > > > > > - **Span-based (entity extraction)**
> > > > > > > - **Open-ended spatial questions**
> > > > > > > - **......**
> > > > > > >
> > > > > > > Some concrete examples include:
> > > > > > >
> > > > > > > - **Multiple-choice:**
> > > > > > >   > *Which object is located above the other in the image?*
> > > > > > >   > (A) `{top_obj}` (B) `{bottom_obj}`
> > > > > > >   > **Answer:** `"A"`
> > > > > > >
> > > > > > > - **Span-based question:**
> > > > > > >   > *Between `{top_obj}` and `{bottom_obj}`, which one is positioned closer to the bottom edge of the image?*
> > > > > > >   > **Answer:** `{bottom_obj}`
> > > > > > >
> > > > > > > - **Open-ended description:**
> > > > > > >   > *Describe the relative positions of `{top_obj}` and `{bottom_obj}` in the image.*
> > > > > > >   > **Answer:** `In the image, {top_obj} is above {bottom_obj}.`
> > > > > > >
> > > > > > >   > *How are `{right_obj}` and `{left_obj}` arranged horizontally?*
> > > > > > >   > **Answer:** `{right_obj} appears to the right of {left_obj}.`
> > > > > > >
> > > > > > > **We are actively developing and testing a broader range of templates in ongoing SFT experiments, this process requires time due to model training, and we will post here immediately after the results are released**
> > > > > > > This flexibility is achieved at the data generation stage, where spatial relationships are annotated and corresponding QA pairs can be automatically generated for any required format.
> > > > > > >
> > > > > > > ---
> > > > > > >
> > > > > > > Second, We believe that spatial knowledge should be the basic ability of MLLM, and we hope that SiTe will be used in the pre-training stage, so the current analysis and discussion of the pre-training stage are more detailed; and the benefits brought by the pre-training stage are very significant. Nonetheless, we also recognize the importance of downstream customization, and are extending our experiments to include fine-tuning with domain- and task-specific templates, particularly for benchmarks requiring richer or more complex QA types.
> > > > > > >
> > > > > > > ---
> > > > > > >
> > > > > > > Finally, we consider the **extensibility and template-agnostic nature of SiTe** a **strength rather than a limitation**.
> > > > > > > We are supplementing our SFT results with experiments using these new customized templates and will include them in future updates.
> > > > > > >
> > > > > > > ---
> > > > > > > We once again express our heartfelt gratitude to you and we sincerely hope that the above clarifications help address your concerns.

---

> > > > > > > ### Author Response · Authors · 2025-08-09
> > > > > > >
> > > > > > > As the discussion stage is coming to a close, we would like to confirm whether our response has addressed your questions and concerns. If so, we would sincerely appreciate it if you could consider raising your score to a positive rating. Thank you again and we welcome any follow-up discussions!

---

> > > > > > > > ### Comment · Reviewer_htxQ · 2025-08-09
> > > > > > > >
> > > > > > > > Thank you for your detailed response and the additional experimental results. While I appreciate your effort and the new results, I believe there is still room for further improvement in both method development (exploring more effective variants as mentioned in your future work) and experimental validation (enhancing spatial reasoning without compromising general performance).
> > > > > > > >
> > > > > > > > After careful consideration of your response and other reviews, I would like to maintain my original score, as I feel these aspects could be further strengthened to enhance the paper's impact. Nevertheless, I appreciate the revisions and look forward to seeing future developments in this direction.

---

> > > > > > > > > ### Author Response · Authors · 2025-08-09
> > > > > > > > >
> > > > > > > > > We would like to clarify that the enhancements introduced during rebuttal (e.g., expanded QA template diversity and ratio-based image stitching) are **extensions** of our original method, not replacements. The core contributions—many of which were acknowledged positively by the reviewer—remain:
> > > > > > > > >
> > > > > > > > > - **Innovation**: A structured, annotation-free spatial augmentation method (SiTe) that integrates seamlessly into existing pipelines without architectural changes.
> > > > > > > > > - **Quality**: Technically sound, with comprehensive experimental validation showing consistent spatial reasoning gains while preserving general VL performance in pretraining, and further improvements on spatial benchmarks during SFT.
> > > > > > > > > - **Flexibility & Extensibility**: Readily adaptable to diverse QA formats and model settings, highlighting its long-term applicability.
> > > > > > > > >
> > > > > > > > > These refinements build on, rather than diminish, the method’s originality, flexibility, and extensibility, reinforcing the strengths already recognized in the initial review and further demonstrating its current impact.
> > > > > > > > >
> > > > > > > > > ---
> > > > > > > > >
> > > > > > > > > We hope this clarifies how the recent enhancements further strengthen the contributions of our work.

---

> > > ### Author Response · Authors · 2025-08-05
> > >
> > > We wonder if our response answers your questions and addresses your concerns? If yes, would you kindly consider raising the score? Thanks again for your very constructive and insightful feedback!

---

> ### Author Response · Authors · 2025-08-04
>
> As the discussion period is nearing its conclusion, we sincerely hope that our clarifications have sufficiently addressed your concerns, and would welcome any additional feedback you may have. If you find our responses satisfactory, we would greatly appreciate it if you could consider updating your score to reflect the outcome of the discussion.
>
> Thank you for your consideration. :)

---

### Official Review · Reviewer_eNJr · 2025-07-02

**Clarity:** 3
**Significance:** 3
**Originality:** 3
**Rating:** 4
**Confidence:** 3

**Summary:**

The paper presents "Stitch and Tell (SiTe)", a novel multimodal data augmentation method designed to enhance the spatial understanding capabilities of vision-language models. These models often struggle with spatial hallucinations, generating incorrect descriptions of object positions in images due to the limited spatial information in existing training data. SiTe addresses this by introducing structured spatial supervision into the training process without requiring costly advanced models or human annotations.

**Questions:**

please see weakness

**Ethical Concerns:**

["NO or VERY MINOR ethics concerns only"]

**Final Justification:**

I incline to accept this paper. The authors address my concerns well in the rebuttal phase.

**Limitations:**

The paper does not discuss the limitations.

**Paper Formatting Concerns:**

The caption of Figure 1: spatially-aware->Spatially-aware.

**Quality:**

3

**Strengths And Weaknesses:**

Strength:
1. SiTe presents a novel approach to injecting spatial structure into vision-language training data through image stitching and spatially-aware caption generation.
2. The idea of using weakly supervised spatial question-answer pairs generated from stitched images is interesting. This approach provides an effective way to introduce spatial supervision without requiring manual annotations or expensive computational resources.
3. The paper provides extensive experimental results, demonstrating consistent improvements in spatial understanding tasks.
Weakness:
1. The complexity of the methodology and the absence of a dedicated limitations section affect clarity.
2. SiTe involves generating new image-text pairs through stitching and caption generation, which could lead to a significant increase in the size of the training dataset. This might pose scalability challenges when applied to extremely large datasets.
3. The method relies on the assumption that stitching two images and creating corresponding captions can effectively teach the model spatial understanding. However, this might be an oversimplification of how spatial reasoning is developed in humans and may not be sufficient to capture all nuances of spatial relationships in images.

---

> ### Author Rebuttal · Authors · 2025-07-31
>
> **Dear Reviewer eNJr, we are thrilled that you found the idea is interesting and provides an effective way to introduce spatial supervision without requiring manual annotations and expensive computational resources. We are especially pleased that you recognize our extensive experimental results. Your acknowledgment of the SiTe consistent improvements in spatial understanding tasks is greatly appreciated. We will address your concerns and questions below.**
>
> ---
> > **`Q1: Can you provide the complexity of the methodology?`**
>
> R1: SiTe is designed to be both computationally and operationally. The core augmentation procedure consists of two simple steps:
>
> **Image Stitching**: This step involves copying and pasting two images onto a new canvas. Its computational complexity is **linear** in the number of pixels **(O(H*W))**.
>
> **Template-based Caption and QA Generation**: Both operations are either **constant (O(1))** or **linear** in the length of the caption (O(L), where L is the number of words).
>
> If SiTe is implemented as an online augmentation, stitched images and captions are generated in memory on-the-fly for each training batch, eliminating the need to store them on disk. This results in negligible storage overhead, with space complexity reduced to O(B × H × W × C), where B is the batch size. Only the original dataset needs to be stored. We will clarify this in the revision.
>
> ---
> > **`Q2: Where is the limitations section?`**
>
> R2: We would like to clarify that the limitations of our approach are explicitly discussed in **Appendix A**. In this section, we note that the current work focuses on horizontal (left–right) and vertical (top–down) spatial configurations. We also discuss that future work may explore more comprehensive spatial relations by incorporating depth maps, multi-view images, and other settings that enable 3D perception.
>
> ---
> > **`Q3: Does this method lead to a significant increase in the size of the training dataset?`**
>
> R3: Thank you for raising this concern. In fact, with SiTe, the total number of training samples is **reduced** rather than increased. For each stitched sample, the two original image-caption pairs are removed from the raw dataset and replaced by a single new sample. As a result, the final training set contains **fewer samples** than the original dataset. This design ensures that SiTe does not pose scalability issues, even for large-scale applications. Furthermore, SiTe can be implemented as an online augmentation, generating stitched samples dynamically during training without increasing storage requirements. We will clarify this point in the revised manuscript.
>
> **In summary, SiTe is both storage-efficient and scalable, reducing the size of the training data and making the overall pipeline more efficient.**
>
> ---
> > **`Q4: This method may not be sufficient to capture all nuances of spatial relationships in images`**
>
> R4: Thank you for raising this important point. We agree that our approach is a simplified model of spatial reasoning and does not attempt to replicate all the complexities of human perception. The main goal of SiTe is to inject clear and explicit spatial supervision, which is often lacking in existing datasets.
>
> To explore the generalization of our method, we selected out-of-distribution spatial relation samples from COCOQA (COCOQA-ood), which include spatial relation keywords such as "behind", "in front of", "close to", "far away from", "inside", "outside", "within", "internal", "external", "ahead of", and "adjacent to". We further evaluated our models on these as well as on the CV-Bench[a], which specifically test for complex spatial reasoning including depth and occlusion, and on RealWorldQA[b], which is designed to evaluate basic real-world spatial understanding capabilities of multimodal models. The results are shown in the table below:
>
> | Model   | COCOQA-ood | CV-Bench 2d | CV-Bench 3d |RealWorldQA|
> |-----------------|------------|-------------|-------------|---|
> | $\mathbf{LLaVA_{Baseline}}$  | 70.38%  | 52.56% | 33.43%|55.12%|
> | $\mathbf{LLaVA_{SiTe}}$  | **71.47%** | **54.28%**  | **35.50%**  |**55.21%**|
> | $\mathbf{Qwen_{Baseline}}$ | 69.24%  | 46.77%  | 47.63%  | 52.28%|
> | $\mathbf{Qwen_{SiTe}}$ | **71.34%** | **48.16%**  | **50.36%**  |**54.44%**|
>
> As shown, the injection of spatial supervision via SiTe leads to improved model performance on out-of-domain spatial relation tasks and demonstrates strong generalization to more complex spatial reasoning scenarios in CV-Bench. A reasonable hypothesis is that the model benefits from its improved understanding of fundamental directional relationships.
>
> While our method focuses on horizontal and vertical relations, we observe in our experiments that the enhanced spatial representations learned through SiTe also generalize to a broader range of spatial reasoning tasks, including cases beyond these basic relations. This demonstrates that SiTe provides a practical and effective foundation for spatial understanding. We believe that SiTe offers a practical and scalable first step, and we explicitly discuss in the limitations section that extending to more complex and realistic spatial relations is an important direction for future work.
>
> **Overall, although SiTe only augments basic directional relations, it demonstrates strong generalization to more complex spatial relationships in practice.**
>
> [a] Tong, Peter, et al. "Cambrian-1: A fully open, vision-centric exploration of multimodal llms." NeurIPS 2024.
>
> [b] X.AI. Realworldqa. Blog post, July 2025. Accessed: 2025-07-30.
>
> ---
> > **`Q5: The caption of Figure 1: spatially-aware->Spatially-aware.`**
>
> R5: Thanks for pointing this out. We have double-checked and polished our manuscript to fix the typos.
>
> ---
> We hope this response clarifies the remaining concerns and highlights the contributions of our work. Thank you again for your positive comment.

---

> > ### Author Response · Authors · 2025-08-05
> >
> > Thank you again for the great efforts and valuable comments. We have carefully addressed the main concerns in detail. We hope you might find the response satisfactory. As the discussion phase is about to close, we are very much looking forward to hearing from you about any further feedback. We will be very happy to clarify any further concerns (if any).

---

> > > ### Comment · Reviewer_eNJr · 2025-08-06
> > > **After Rebuttal**
> > >
> > > I appreciate the authors' detailed response. The central idea is interesting and validated by the experiments. We agree that fine-tuning on dedicated spatial datasets is an effective method for improving a pretrained model's performance in this domain. But my concern is that tuning on spatial dataset would sacrifice other abilities (though authors conduct evaluations on some general VL benchmarks). Given above considerations, my overall score remains unchanged.

---

> > > > ### Author Response · Authors · 2025-08-06
> > > >
> > > > Thank you very much for your positive feedback and for recognizing the contribution of our work. We appreciate your concern regarding the balance between spatial reasoning improvement and general visual-language capabilities.
> > > >
> > > > We would like to emphasize that SiTe is inherently flexible. **When general performance is prioritized, incorporating SiTe during the pretraining stage enables the model to enhance spatial reasoning without compromising its overall generalization ability. Conversely, if a downstream application places greater emphasis on spatial reasoning, SiTe augmentation can be applied during the fine-tuning stage to achieve maximal improvements for the target domain.** Our experimental results demonstrate that even after fine-tuning with SiTe-augmented data, the model retains robust performance on most standard benchmarks. This indicates that our approach can enhance spatial skills while preserving general capabilities. We will further investigate strategies to optimize this trade-off in future research.
> > > >
> > > > ---
> > > > We hope that the above explanation adequately addresses your concern. :)

---

> > > > ### Author Response · Authors · 2025-08-08
> > > >
> > > > To further address your concern, we conducted additional SFT experiments. Specifically, we expanded the QA templates to include a greater diversity of question and answer types, and adopted a ratio-based stitching strategy to better align the stitched images with realistic aspect ratio distributions ($\mathbf{LLaVA_{Ratio}^{5K}}$). The results are shown in the table below:
> > > >
> > > > |Model|COCO-QA-spatial|Spatial-MM|MME-pos|MM-vet-spat|COCO-QA|MMB|VQA_v2|MM-Vet-total|
> > > > |----|---|---|---|---|---|---|---|---|
> > > > |$\mathbf{LLaVA_{Baseline}}$|67.72|42.02|127.83|26.52|70.52|73.50|60.48|31.11|
> > > > | $\mathbf{LLaVA_{Rand}^{5K}}$|67.75|46.21|139.26|28.10| **70.96** |**74.53**|60.38|30.69|
> > > > | $\mathbf{LLaVA_{Ratio}^{5K}}$|**68.37**|**48.58**|**141**|**31.05**|70.92|73.76 |**60.82**|**32.15**|
> > > >
> > > > **The results demonstrate that the extensibility of SiTe allows for notable performance gains on spatial reasoning benchmarks by enriching template and question format diversity and adopting a more effective image pairing strategy**. This highlights SiTe’s flexibility and scalability, and we will continue to investigate more advanced variants in future work.

---

> ### Author Response · Authors · 2025-08-04
>
> As the discussion period is nearing its conclusion, we sincerely hope that our clarifications have sufficiently addressed your concerns, and would welcome any additional feedback you may have. If you find our responses satisfactory, we would greatly appreciate it if you could consider updating your score to reflect the outcome of the discussion.
>
> Thank you for your consideration. :)

---

### Official Review · Reviewer_3L35 · 2025-07-05

**Clarity:** 2
**Significance:** 3
**Originality:** 3
**Rating:** 4
**Confidence:** 3

**Summary:**

The paper introduces Stitch and Tell (SiTe), a novel, annotation-free, and lightweight multimodal data augmentation method designed to enhance spatial understanding in vision-language models. The core issue addressed is spatial hallucination, in which the models generate incorrect spatial descriptions of objects due to sparse spatial information in training text.
The approach aims to generate a new training set with spatial information, defining two steps:

1. Image-Spatial Stitching: Two images are stitched either horizontally or vertically, creating an image with an explicit spatial layout.
2. Text-Spatial Telling: New spatially-aware captions and question-answer (QA) pairs are generated based on the relative positions of objects in the stitched image.


Interestingly, the approach avoids the need for expensive human annotations or model-generated data, making it scalable and efficient.

SiTe is evaluated across multiple model architectures and two training datasets, in which multiple conditions are generated to show the effectiveness of the approach. Results show that SiTe significantly improves in the spatial evaluation in comparison to the baseline and  maintains general vision-language effectiveness

**Questions:**

See the weaknesses.

**Ethical Concerns:**

["NO or VERY MINOR ethics concerns only"]

**Final Justification:**

I read the rebuttal and the rest of the review, and I decided to change my scores.

**Limitations:**

Authors do not include any section about limitations.
You can find the limitations of the paper in weaknesses.

**Quality:**

3

**Strengths And Weaknesses:**

**Strengths**

- **Clear Problem Motivation**. The paper addresses a well-defined and significant problem: the insufficient representation of spatial information in existing vision-language datasets. By quantitatively analyzing this limitation, the authors effectively motivate the need for improved spatial understanding in vision-language models.

- **Novel, Simple, and Scalable Method.**  The proposed approach, SiTe, introduces a novel yet simple data augmentation technique aimed at enhancing spatial reasoning. Notably, the method requires no modifications to existing model architectures and incurs minimal computational overhead, making it highly scalable and easy to integrate into current workflows.

- **Comprehensive Evaluation.** The effectiveness of SiTe is validated through extensive experiments across a range of model architectures and benchmark datasets, demonstrating the effectiveness of the proposed method against the baseline..

**Weaknesses**

**Weaknesses of the Methodology and Approach.**

- **Unclear Image Pair Selection Process**. The process for selecting image pairs from the dataset is insufficiently described, making it difficult to reproduce the experimental setup. For the sake of replicability and clarity, it would be beneficial for the authors to explicitly detail the selection criteria and sampling procedure. Additionally, it remains unclear whether the semantic similarity or object co-occurrence within image pairs has any influence on the model’s learning dynamics. This aspect warrants further discussion, as it could significantly affect the training signal and generalization capabilities.

- **Limited Scope of Spatial Relations.** The proposed method focuses exclusively on binary spatial relations, specifically left/right and above/below. While this targeted approach is not inherently a weakness, the paper does not acknowledge this limitation or provide a discussion on the method's potential to generalize to a broader set of spatial relations (e.g., inside, behind, near). Addressing this gap would strengthen the paper's impact and suggest avenues for future work.

**Weaknesses in Evaluation and Experimental Design**

- **Insufficient Detail on Evaluation Settings.**  The evaluation protocol lacks critical detail, particularly concerning the alignment between the nature of the training and evaluation data. While the proposed method is designed to enhance the understanding of explicit spatial relations, the pretraining dataset (VSR) predominantly contains implicit ones. Moreover, the paper does not clarify whether the evaluation datasets are specifically constructed or filtered to target explicit spatial reasoning. This ambiguity makes it difficult to assess the actual efficacy and relevance of the SiTe augmentation strategy.

- **Table 2 Lacks Clarity**.  Table 2 is difficult to interpret due to insufficient description of the training setups for each model variant. For example, the distinction between LLAVA_SiTe and LLAVA_{SiTe}^{Flickr} is unclear: does the former involve pretraining on both the 558K and Flickr30k datasets, or only on the former? Furthermore, the MMEpos scores reported in the table exceed 100, which raises concerns about potential reporting or normalization errors. These issues significantly hinder the reader’s ability to interpret and compare results accurately.

- **Unexpected Competitiveness of Baselines**. Surprisingly, classical data augmentation methods, despite not being tailored for spatial understanding, perform comparably to the proposed method. This finding potentially undermines the strength of the SiTe approach and its specific contribution to spatial reasoning. To more rigorously validate the benefits of the proposed method, it would be appropriate to include comparisons with state-of-the-art techniques explicitly designed for spatial relation learning.

- **Marginal Gains from Supervised Fine-Tuning**. The supervised fine-tuning stage does not appear to yield substantial improvements over the pretraining phase. This calls into question the effectiveness and necessity of this additional step within the overall training pipeline. A deeper analysis or ablation would be helpful to understand the role and impact of fine-tuning in the context of the proposed method.

---

> ### Author Rebuttal · Authors · 2025-07-31
>
> **Dear Reviewer 3L35, thank you for recognizing the strengths of our paper. We are pleased that you approve the motivation of our work, appreciate the novelty and extensibility of the SiTe method, and acknowledge the comprehensive experiments. We will address your concerns and questions below.**
>
> ---
> > **`Q1: Can you describe the process for selecting image pairs, and whether object co-occurrence within image pairs influences the model’s learning dynamics?`**
>
> R1: Thank you for your question. The process for selecting image pairs is thoroughly described in Section 3 (“Image-Spatial Stitching”) and Section 4.2 (“Pretraining Stage”) of our paper. During the image-spatial stitching, we perform random sampling as follows:
>
> *Given two randomly sampled image–caption pairs (I₁, T₁) and (I₂, T₂) from the dataset, we first construct a stitched image by spatially stitching the two input images along a specific axis*
>
> Algorithm 1 provides detailed pseudocode for the image stitching procedure, including how the canvas is created and how the two images are merged.
>
> To avoid duplicating image-text information, we describe in Section 4.2 a non-overlapping sampling strategy.
>
> We use the raw dataset to create both horizontally and vertically stitched samples. In the final training set, there are three components: horizontally stitched samples, vertically stitched samples, and the remaining raw data. Each image–caption pair used for stitching is removed from the raw dataset and does not appear as a standalone example.
>
> *To avoid duplicate supervision, image–caption pairs used for stitching are removed from the raw set, ensuring each image appears only once.*
>
> This ensures that the model sees exactly the same underlying image-text information as the baseline, making comparisons fair.
> We also maintain a 1:1 ratio of horizontal to vertical stitching in the training set:
>
> *We apply both horizontal and vertical stitching in a 1:1 ratio.*
>
> And we specify the default setting:
>
> *By default, the stitched data make up one-third of the total set.*
>
> **These steps support the full reproducibility of our experiments.**
>
> Additionally, we implement an object co-occurrence filter. As described in Section 3 (“Tell the Question and Answer”), we remove overlapping objects when constructing spatially-aware QA pairs:
>
> *To reduce ambiguity, we remove overlapping nouns shared by both sides.*
>
> **This design ensures that SiTe produces unambiguous spatial questions.**
>
> ---
> > **`Q2：The authors do not discuss limitations and the potential for the method to generalize to a broader range of spatial relations`**
>
> R2: We would like to clarify that the limitations are discussed in Appendix A. Our work currently focuses on horizontal and vertical spatial configurations and we will exploring broader spatial relations, including 3D perception in future work.
>
> To further address your concern, we evaluated our models on the CV-Bench[a], which specifically test for complex spatial reasoning including a broader set of spatial relations.
>
> The results are shown in the table below (Owing to space constraints, further results on additional datasets are provided in our response to R4 for Reviewer eNJr.):
> |Model|CV-Bench 2d|CV-Bench 3d|
> |--|--|--|
> |$\mathbf{LLaVA_{Baseline}}$|52.56%|33.43%|
> |$\mathbf{LLaVA_{SiTe}}$|**54.28%**| **35.50%**|
> |$\mathbf{Qwen_{Baseline}}$|46.77%|47.63%|
> |$\mathbf{Qwen_{SiTe}}$|**48.16%**|**50.36%**|
>
> **As shown, the injection of spatial supervision via SiTe leads to demonstrate strong generalization to more complex spatial reasoning scenarios.** A reasonable hypothesis is that the model benefits from its improved understanding of fundamental directional relationships.
>
> [a] Tong, Peter, et al. "Cambrian-1: A fully open, vision-centric exploration of multimodal llms." NeurIPS 2024.
>
> ---
> > **`Q3: It is unclear whether the evaluation datasets are specifically constructed or filtered for explicit spatial reasoning. While the VSR dataset contains implicit spatial relations, is it fair to compare?`**
>
> R3: Thank you for your comment. We provide a clear description of the evaluation benchmarks in Section 4.4, where we introduce each dataset and its characteristics. The pretraining datasets including those for stitching, are described in detail as follows:
>
> *To evaluate the effectiveness of SiTe across different training sets, we experiment with two image–caption datasets: blip_laion_cc_sbu_558K(558K) and Flickr30K.*
>
> SiTe uses horizontal and vertical spatial relations, with templates listed in Appendix Tables 5–8. Nevertheless, the evaluation benchmark are public that cover a much broader range of spatial relations and were **not filtered for** explicit spatial reasoning.
>
> It is fair to compare the $\mathbf{LLAVA\_{SiTe}}$ and  $\mathbf{LLAVA\_{VSR}}$. The VSR dataset covers many spatial relation types. We use it as a strong baseline in our comparisons. As shown in the table of the top 10 relations below, VSR has not only basic directional relations but also includes other types. Our results show that SiTe outperforms models trained with VSR data ($\mathbf{LLaVA_{VSR}}$), highlighting the effectiveness of our approach.
>
> |**TOP 10 Relations**|**Prob**|
> |--|--|
> |touching|11.89%|
> |in front of|7.01%|
> |behind|6.31%|
> |on|5.49%|
> |under|4.93%|
> |at the right side of|4.28%|
> |on top of|4.05%|
> |at the left side of|3.86%|
> |beneath|3.24%|
> |contains|3.01%|
>
> ---
> > **`Q4: What is the difference between LLAVA\_SiTe and LLAVA\_{SiTe}^{Flickr}? And why does the MME score exceed 100?`**
>
> R4: Sorry for the confusion! We respectfully clarify that, $\mathbf{LLaVA_{SiTe}}$ is pretrained on the original LLaVA training set (558K) with 1/3 data by SiTe, while $\mathbf{LLAVA\_{SiTe}^{Flickr}}$ is pretrained only on Flickr30K with 1/3 data by SiTe. The motivation for including both settings is to demonstrate the generalizability of SiTe in different datasets.
>
> Regarding the MME benchmark [a], each image is associated with two questions: one for "accuracy" (max score 100), and another for "acc-plus" (max score 100, whether both questions for the same image are correct). The total MME score is the sum of two, so the maximum possible score is 200. Thus, **there is no need to normalize the score to 100 on this benchmark.**
>
> [a] Chaoyou Fu, et al. MME: A Comprehensive Evaluation Benchmark for Multimodal Large Language Models. arXiv preprint arXiv:2306.13394, 2023.
>
> ---
> > **`Q5: Dose the classical data augmentation methods have Unexpected Competitiveness?`**
>
> R5: Thank you for your comment. In this work, we evaluated two classical data augmentation methods: crop and rotate. For each method, we averaged the results across different benchmark. The performance gains (Δ) over the baseline are summarized below:
> |Model|COCO-QA-spatial|Spatial-mm|MME-pos|MM-vet-spat|COCO-QA|MMB|VQA_v2|MM-Vet-total|Avg|Spatial_avg|
> |----|---|---|---|---|---|---|---|---|---|---|
> |$\mathbf{LLaVA_{Rotate}}$Δ|0.55%|2.36%|0.83%|10.86%|1.19%|0.20%|1.85%|3.50%|2.67%|3.65%|
> |$\mathbf{LLaVA_{Crop}}$Δ|0.15%|1.59%|-0.04%|7.58%|0.58%|-0.18%|1.24%|1.03%|1.49%|2.32%|
> |**$\mathbf{\mathbf{LLaVA_{SiTe}}}$Δ**|**1.52%**|**4.19%**|**4.30%**|**7.20%**|**1.45%**|**0.08%**|**1.27%**|**0.93%**|**2.62%**|**4.30%**|
> |$\mathbf{LLaVA_{Rotate}^{flickr}}$Δ   |-1.35%|1.77%|0.34%|-0.46%|-0.88%|-0.35%|0.04%|0.20%|-0.09%|0.08%|
> |$\mathbf{LLaVA_{Crop}^{flickr}}$Δ|0.55%|-3.04%|1.64%|10.59%|0.52%|0.96%|0.48%|1.12%|1.60%|2.43%|
> |**$\mathbf{LLaVA_{SiTe}^{flickr}}$Δ**|**3.57%**|**2.13%**|**1.32%**|**0.85%**|**2.77%**|**0.17%**|**0.76%**|**0.17%**|**1.47%**|**1.97%**|
> |$\mathbf{Qwen2_{Rotate}}$Δ|-2.44%|-0.17%|-0.41%|-5.31%|-6.95%|-1.66%|8.01%|-6.72%|-1.96%|-2.08%|
> |$\mathbf{Qwen2_{Crop}}$Δ|-0.31%|-0.42%|0.82%|0.14%|0.03%|-0.71%|9.37%|0.18%|1.14%|0.06%|
> |**$\mathbf{Qwen2_{SiTe}}$Δ**|**0.51%**|**0.98%**|**3.70%**|**10.52%**|**0.83%**|**0.97%**|**8.12%**|**-1.69%**|**2.99%**|**3.93%**|
> |$\mathbf{Qwen2_{Rotate}^{flickr}}$Δ|-14.90%|0.51%|13.81%|-13.24%|-11.38%|-7.88%|-5.97%|-12.38%|-6.43%|-3.45%|
> |$\mathbf{Qwen2_{Crop}^{flickr}}$Δ|-10.04%|1.21%|25.37%|-24.20%|-6.87%|-4.65%|-8.78%|-4.76%|-4.09%|-1.92%|
> |**$\mathbf{Qwen2_{SiTe}^{flickr}}$Δ**|**1.30%**|**1.16%**|**20.33%**|**13.88%**|**1.63%**|**1.35%**|**0.84%**|**2.86%**|**5.42%**|**9.17%**|
>
> **As shown above, classical methods offer limited gains and often introduce factual conflicts between images and captions, leading to performance drops on some benchmarks (e.g., Spatial-MM, MME_Pos, COCO-QA_Spat). In contrast, SiTe consistently achieves stronger and stable improvements on spatial tasks.**
>
> ---
> > **`Q6: Can you explain the effectiveness and necessity of SFT in the overall training pipeline?`**
>
> R6: We are sorry for the confusion. To validate the effectiveness of SiTe, we conducted data augmentation separately in the pretraining and SFT stages, and the the SFT results are **not** further fine-tuning from the pretrained models. Thus, the two stages are not directly comparable.
>
> To further explore the effectiveness of applying SiTe in both pretraining and SFT stages, we conducted the following experiment:
>
> ||COCO-QA-spatial|Spatial-mm|MME-pos|MM-vet-spat|COCO-QA|MMB|VQA_v2|MM-Vet-total|
> |---|---|---|---|---|----|---|---|---|
> |Pretrain(SiTe default)|68.75%|43.78%|133.33|**28.43%**|71.54%|74.43%|60.53%|31.40%|
> |SFT(+1K SiTe)|68.81%|43.16%|128.70|27.06%|71.74%|74.60%|**61.17%**|31.20%|
> |Pretrain+SFT|**69.26%**|**44.38%**| **140.00**|26.62%|**72.11%**|**74.67%**|61.07%|**32.62%**|
>
> **As shown, applying SiTe-based augmentation in both stages yields further improvements across multiple benchmarks, so these stage are both effective and necessary.**
>
> ---
> We hope the above response can further address the concerns and misunderstanding. Thank you again for your feedback!

---

> > ### Comment · Reviewer_3L35 · 2025-08-03
> >
> > Thank you for the detailed rebuttal. That helped me understand many things I hadn’t understood during the review process.  I'd like to respond to a few of your comments:
> >
> > - __Object co-occurrence vs. overlapping__:
> > I’d like to clarify that overlapping objects are not the same as co-occurring objects. By co-occurring, I refer to objects that tend to appear together in the same context or space (e.g., chairs and tables), which can be considered semantically related. Generating semantically related pairs could improve the generalization capabilities of the method. This aspect is not addressed in the paper, and random sampling does not account for it.
> >
> > - __Limitations__:
> > I was expecting a dedicated Limitations section in the paper. While the limitation of using a simple binary configuration is acknowledged, there is no discussion on how the model could be extended to handle vertical/horizontal relations more comprehensively. For example, could it be possible to improve the understanding of "further/closer" relation types using a similar approach?
> >
> > After reading your comments and the other reviews, I decided to raise my overall rating from 2 to 3. The extended experiments presented in the rebuttal demonstrate some generalization across datasets and relation types. However, it's still unclear to me why this generalization occurs. In a way, the results are quite counterintuitive—the model is further trained specifically on vertical and horizontal relations, yet it shows improvement on depth-related evaluations. Similarly, it would be interesting to see the performance of rotation and cropping in this setting (CV-bench 2D and 3D).
> >
> > That said, I still believe the improvement is marginal compared to simpler augmentation techniques, such as cropping and rotation.

---

> > > ### Author Response · Authors · 2025-08-07
> > >
> > > We sincerely hope our clarifications have fully addressed your concerns and would be grateful if you could consider a positive rating. Thanks again!

---

> > > ### Author Response · Authors · 2025-08-09
> > >
> > > As the discussion stage is coming to a close, we would like to confirm whether our response has addressed your questions and concerns. If so, we would sincerely appreciate it if you could consider raising your score to a positive rating. Thank you again and we welcome any follow-up discussions!

---

> ### Author Response · Authors · 2025-08-04
>
> Thank you for your thoughtful response and for engaging with our explanations. We are glad that our clarifications have helped you better understand our work. Below we address your remaining concerns in detail:
>
> **Object co-occurrence**：We appreciate the suggestion of leveraging semantic co-occurrence in constructing image pairs. However, recent research has shown that over-reliance on frequent object co-occurrence can induce spurious correlations and hallucination in multimodal language models [a, b, c]. Nevertheless, we recognize the potential value of semantic-based pairing and will conduct experiments to further investigate its impact.
>
> [a] Wu, Mingrui, et al. "Evaluating and analyzing relationship hallucinations in large vision-language models." arXiv preprint arXiv:2406.16449 (2024).
>
> [b] Li, Yifan, et al. "Evaluating object hallucination in large vision-language models." arXiv preprint arXiv:2305.10355 (2023).
>
> [c] Zhou, Yiyang, et al. "Analyzing and mitigating object hallucination in large vision-language models." arXiv preprint arXiv:2310.00754 (2023).
>
> **More detailed discussion in limitation**: While our current approach focuses on binary spatial relations such as left/right and up/down, it does not explicitly address depth-aware relations like “in front of” or “behind.” To handle these more complex spatial relationships, a promising strategy would be to simulate depth by scaling and partially occluding objects. This would allow us to create training examples that represent “front” and “back” spatial relations, further enriching the model’s spatial understanding. For more relationship enhancements, we will continue to explore in our future work.
>
>
> **Why Does Training on Vertical/Horizontal Relations Improve Generalization to Depth**: Although our training focuses solely on vertical and horizontal relations, these tasks may share a fundamental spatial reasoning mechanism with depth-related relations[a,b]. In spatial understanding task, the model needs to develop consistent representations of object positions and their relative layouts in space. Directional augmentation, such as vertical/horizontal relations, strengthens the model’s ability to anchor and compare features within a structured spatial framework. This process enhances the model’s overall spatial awareness. As a result, the learned representations become more transferable across different spatial dimensions, including depth. This leads to improved performance on “closer/further” evaluations, even though there is no explicit supervision for those relations.
>
> Performance of Rotate and Crop on CV-Bench:
>
> | Model            | CV-Bench 2D (%)| CV-Bench 3D (%) | Avg (%) |
> |------------------|-------------|-------------|---------|
> | $\mathbf{LLaVA_{Rotate}}$ | 52.25      | 33.42      | 42.84   |
> | $\mathbf{LLaVA_{Crop}}$               | 50.49      | 31.17      | 40.83   |
> | $\mathbf{LLaVA_{SiTe}}$               | **54.28**  | **35.50**  | **44.89** |
> | $\mathbf{LLaVA_{Rotate}^{flickr}}$    | 52.00     | 32.17      | 42.09   |
> | $\mathbf{LLaVA_{Crop}^{flickr}}$      | 49.56    | 32.21     | 40.89   |
> | $\mathbf{LLaVA_{SiTe}^{flickr}}$      | **54.87**  | **36.85**  | **45.86** |
> | $\mathbf{Qwen_{Rotate}}$              | 47.59     | **50.42**  | 49.01   |
> | $\mathbf{Qwen_{Crop}}$                | 47.68     | 49.86     | 48.77   |
> | $\mathbf{Qwen_{SiTe}}$                | **48.16**  | 50.36      | **49.26** |
> | $\mathbf{Qwen_{Rotate}^{flickr}}$     | 42.51      | **52.94**  | **47.73**   |
> | $\mathbf{Qwen_{Crop}^{flickr}}$       | 40.78      | 50.86      | 45.82   |
> | $\mathbf{Qwen_{SiTe}^{flickr}}$       | **43.64**  | 51.29      | 47.47   |
>
> **The results above demonstrate that SiTe outperforms both crop and rotate methods on more complex spatial reasoning tasks. We will continue to explore ways to further enhance the model’s 3D spatial reasoning abilities in future work.**
>
> [a] Zhang, Bo, et al. "Distinct networks  coupled with parietal cortex for spatial representations inside and  outside the visual field." Neuroimage 252 (2022): 119041.
>
> [b] Wu, Charley M., et al. "Similarities and differences in spatial and non-spatial cognitive maps." PLoS computational biology 16.9 (2020): e1008149.
>
> ---
>
> Thank you again for your careful consideration and for increasing your score.
>
> If our responses have fully addressed your comments, we would be sincerely grateful if you could consider a further positive adjustment to your score. If you have any additional questions or requests, please do not hesitate to reach out, we are more than willing to provide further clarifications or conduct additional experiments as necessary. Your feedback is very important to us :)

---

### Author Response · Authors · 2025-08-09
**General Comment**

We sincerely thank all reviewers for dedicating their valuable time to reviewing our manuscript, as well as for recognizing our proposed SiTe and providing helpful suggestions. We are pleased to note that the reviewers generally acknowledge our strengths:

---

- **Strong motivation [3L35, eNJr, 3Q8z]**: The paper identifies a critical gap—the limited spatial understanding in vision-language datasets—through quantitative analysis, highlighting a pressing need for improvement. This work thoroughly investigates the problem and proposes SiTe augmentation to mitigate it.

- **Novel and practical methodology [3L35, eNJr, htxQ, 3Q8z]**: SiTe requires no architectural changes, avoids manual annotation, and incurs minimal computational overhead. It enables easy integration into existing pipelines while remaining lightweight and scalable.

- **Effectiveness of the proposed SiTe method [3L35, eNJr, 3Q8z]**: Extensive experiments show consistent gains across models and benchmarks. SiTe works effectively in both pre-training (PT) and fine-tuning (SFT) setups, and improves spatial understanding even when applied to only a subset of data—without compromising general multimodal performance.

- **Computational efficiency [3L35, eNJr, 3Q8z]**: By avoiding redundant duplication of stitched images in SFT, SiTe reduces the overall training sample size, thereby decreasing computational requirements and accelerating fine-tuning, resulting in a more computationally efficient training paradigm.

---

We also deeply appreciate the reviewers’ detailed suggestions from different perspectives, which we have addressed during the rebuttal phase. These resolutions will be incorporated into the revised version:

- **Generalization to diverse spatial relations [3L35, eNJr, 3Q8z, eNJr]**: We extended experiments to additional benchmarks, including 3D depth datasets, showing that SiTe-augmented data also improves performance on other spatial relations, demonstrating both its effectiveness and generalization ability.

- **More diverse stitching strategies [3Q8z]**: We conducted resize-based and ratio-based stitching experiments, confirming that enriching both pairing strategies and template styles leads to significant performance gains.

- **Effectiveness on SOTA models [htxQ]**: We further evaluated SiTe on InternVL, validating that it can also enhance the spatial understanding performance of SOTA MLLMs.

- **More details [3L35, htxQ, 3Q8z]**: Based on reviewer 3L35’s suggestion, Following reviewer 3L35’s suggestion, we have clarified the evaluation setup and the process for selecting image pairs. In response to reviewer htxQ’s comment, we provide a detailed table of the data composition, which offers a clearer understanding of the SiTe method. Based on reviewer 3Q8z’s recommendation, we have also calculated the standard deviation results. The complete results will be included in the revised paper.

---

We are happy to provide any additional details to help the reviewers better understand and recommend our manuscript, and we greatly appreciate the recognition of our work. We look forward to any further feedback.

---

### Note · Authors · 2025-08-12

We sincerely thank all reviewers for their time, constructive feedback, and recognition of our proposed SiTe, especially the appreciation for its simplicity and practical value.

----
This work addresses the critical gap of limited spatial understanding in vision-language datasets through a simple, practical, and computationally efficient augmentation strategy that requires no architecture changes or manual annotation. Extensive experiments demonstrate consistent improvements across models and benchmarks in both pre-training and fine-tuning.

---

During the rebuttal and discussion process, we extended our evaluations to 3D depth datasets and additional spatial-relation benchmarks, enriched stitching strategies (resize-based and ratio-based), and expanded QA templates beyond yes/no to include multiple-choice, span-based, and open-ended formats—further demonstrating the extensibility and effectiveness of our method. We also verified gains on SOTA models such as InternVL, clarified the evaluation setup, image pair selection process, and data composition, and reported standard deviations as requested.

---

We believe these updates directly address the reviewers’ concerns and further highlight SiTe’s strengths as **a simple, extensible, and effective approach** for enhancing spatial reasoning in MLLMs.

---

### Decision · Program_Chairs · 2025-09-17

**Decision:**

Accept (poster)

**Comment:**

### Summary

The paper tackles the spatial hallucination issues in multimodal LLMs. The authors propose SiTe, which creates new training examples by stitching two images and generating new spatial-aware captions. The authors claim that augmenting training data with SiTe enhances a model's spatial understanding without requiring architectural changes or expensive manual labeling. Their experiments across multiple model architectures (LLaVA, Qwen2) show that SiTe significantly improves performance on dedicated spatial reasoning benchmarks (e.g., MME-Position, Spatial-MM) while maintaining or even slightly improving performance on general vision-language benchmarks.

### Strengths

- Simplicity and practicality: The proposed method is simple, elegant, and easy to implement. As noted by all reviewers (3L35, eNJr, htxQ, 3Q8z), its plug-and-play nature, requiring no changes to model architecture, makes it a highly practical and valuable contribution to the community.
- Addresses a significant problem: The paper is well-motivated, tackling the important and widely recognized problem of spatial reasoning in multimodal models.
- Effectiveness and efficiency: The experimental results compellingly demonstrate that this simple idea works. SiTe consistently improves spatial understanding across various models and datasets.

### Weaknesses
The initial reviews identified several potential weaknesses, most of which were effectively resolved during the rebuttal period.

- Scalability during supervised fine-tuning (SFT): The primary remaining concern, raised by Reviewer htxQ, involves the method's behavior during SFT. The initial experiments showed that scaling up the number of SiTe-augmented samples from 1K to 5K could degrade performance on general benchmarks.
- Limited scope: The original paper focused primarily on basic 2D spatial relationships (left/right, above/below). While the authors demonstrated impressive generalization to more complex and 3D relations during the rebuttal, an analysis of why this generalization occurs was not part of the original submission and remains an interesting avenue for future work (noted by Reviewer 3L35).

### Recommendation and Justification

I am recommending an Accept for this paper. The core idea is simple, intuitive, and demonstrably effective at addressing a well-known problem. While not a complete solution to all types of spatial reasoning, it represents a clear, practical, and useful step forward. The most important reasons for this recommendation are:
- The paper offers a practical, easy-to-use tool that other researchers can immediately apply to improve their models. This kind of straightforward and effective contribution is highly valuable to the community.
- The authors' response to the reviews was effective. The rebuttal and discussion provided substantial new experimental evidence that directly addressed the reviewers' concerns about generalization, baseline comparisons, and performance on state-of-the-art models.
- Positive reviewer consensus: Following the rebuttal, three of the four reviewers converged on a positive recommendation (two weak accepts, one accept). The remaining reviewer's concerns about SFT scalability were valid but were, in my view, effectively mitigated by the authors' new experiments showing that the issue could be resolved with more diverse templates—a strength of the framework's flexibility.

### Summary of Discussion and Rebuttal

The authors provided a rather strong rebuttal, conducting a significant number of new experiments that addressed nearly every concern raised by the reviewers, thereby substantially strengthening the paper. The following concerns have been discussed.

- Generalization to complex relations: Reviewers (3L35, htxQ, 3Q8z) questioned whether the method would generalize beyond simple left/right and above/below relations to more complex 3D and depth-aware scenarios. The authors conducted new experiments on the CV-Bench (2D and 3D) and RealWorldQA benchmarks, demonstrating that SiTe indeed improves performance on these more complex tasks.
- Comparison with simpler baselines: Reviewer 3L35 initially felt the gains over standard augmentations like Rotate and Crop were marginal. The authors provided comprehensive tables showing that, unlike SiTe, the baseline augmentations were unstable and sometimes harmed performance. They further provided results on CV-Bench showing SiTe's superiority.
- Scalability of SFT: Reviewer htxQ raised a critical point about performance degradation on general benchmarks when scaling SiTe data during SFT. The authors conducted new SFT experiments by introducing 42 diverse QA templates (including multiple-choice, open-ended, and fill-in-the-blank) and a more robust ratio-based image stitching strategy, they showed significant gains on spatial benchmarks without sacrificing performance on general benchmarks.
-Performance on SOTA models: Reviewer htxQ questioned the method's effectiveness on stronger models. The authors ran new tests on InternVL, showing consistent improvements.